# Structural and Functional Strategies in *Cenchrus* Species to Combat Environmental Extremities Imposed by Multiple Abiotic Stresses

**DOI:** 10.3390/plants13020203

**Published:** 2024-01-11

**Authors:** Sana Basharat, Farooq Ahmad, Mansoor Hameed, Muhammad Sajid Aqeel Ahmad, Ansa Asghar, Sana Fatima, Khawaja Shafique Ahmad, Syed Mohsan Raza Shah, Abeer Hashem, Graciela Dolores Avila-Quezada, Elsayed Fathi Abd_Allah, Zaheer Abbas

**Affiliations:** 1Department of Botany, University of Agriculture Faisalabad, Faisalabad 38040, Pakistan; sanabasharat143@gmail.com (S.B.); f.ahmad@uaf.edu.pk (F.A.); mansoor.hameed@uaf.edu.pk (M.H.); sajidakeel@uaf.edu.pk (M.S.A.A.); ansaasghar1122@gmail.com (A.A.); 2Department of Botany, The Government Sadiq College Women University, Bahawalpur 63100, Pakistan; sana.fatima@gscwu.edu.pk; 3Department of Botany, University of Poonch Rawalakot, Rawalakot 12350, Azad Jammu and Kashmir, Pakistan; ahmadks@upr.edu.pk; 4Department of Botany, Division of Science and Technology, University of Education, Lahore 54700, Pakistan; syed.mohsan@ue.edu.pk; 5Botany and Microbiology Department, College of Science, King Saud University, P.O. Box 2460, Riyadh 1451, Saudi Arabia; habeer@ksu.edu.sa; 6Facultad de Ciencias Agrotecnológicas, Universidad Autónoma de Chihuahua, Chihuahua 31350, Mexico; gdavila@uach.mx; 7Plant Production Department, College of Food and Agricultural Sciences, King Saud University, P.O. Box 2460, Riyadh 11451, Saudi Arabia; eabdallah@ksu.edu.sa

**Keywords:** *Cenchrus* spp., organic osmolytes, ionic content, parenchyma, sclerenchyma

## Abstract

Multiple abiotic stresses such as drought, salinity, heat, and cold stress prevailing in natural habitats affect plant growth and development. Different species modify their structural and functional traits to combat these abiotic stresses while growing in stressful environments. *Cenchrus* species, i.e., *Cenchrus pennisetiformis*, *C. setiger*, and *C. prieurii* are widely distributed grasses found growing all over the world. Samples from natural populations were collected from different ecological regions in the Punjab and Khyber Pakhtoonkhwa that were exposed to aridity, salinity, and cold, while one site was designated as normal control. In the present study, structural and functional modifications of three *Cenchrus* species under abiotic stresses were evaluated. It was expected that each *Cenchrus* species may evolve different strategies to cope with multiple abiotic stresses. All *Cenchrus* species responded differently whether growing in normal environment or stressful conditions. The most remarkable feature for survival in *C. pennisetiformis* under cold stress was increased inflorescence and increased stem and root lignification. *C. prieurii* showed better tolerance to saline and cold environments. *C. setiger* showed better development of leaf sheath anatomical traits. The structural and functional modifications in *Cenchrus* species such as development of mechanical tissues provided structural support, while dermal and parenchymatous tissues increased water storage capacity and minimized water loss. An increase in the concentration of organic osmolytes and ionic content aids turgor pressure maintenance and ionic content crucial for plant growth and development. It was concluded that structural and functional alterations in all *Cenchrus* species were very specific and critical for survival under different environmental stresses. The ecological fitness of these species relied on maintenance of growth and biomass production, and the development of mechanical, vascular, dermal and parenchyma tissues under stressful environmental conditions. Moreover, accumulation of beneficial ions (K^+^ and Ca^2+^) and organic osmolytes were critical in turgor maintenance, hence survival of *Cenchrus* spp.

## 1. Introduction

Abiotic stresses severely affect world-wide agriculture. Examples such as drought, salinity, heat, and cold stresses may decrease the yield in an amount of more than 50% in many crops [1,2]. Plants withstand abiotic stresses by modifying their structural and functional traits to survive in a changing climatic condition [3]. Physiological, biochemical, and molecular modifications in response to abiotic stresses may be caused by the increased production of osmolytes, decreased photosynthesis rate, stomatal closure, and induction of stress responsive genes [4].

Multiple abiotic stresses have harmful effects on plant growth and development. Different abiotic stresses occur simultaneously, i.e., salinity and temperature stress [5], drought and temperature [6], heavy metals and temperature that have more severe effects on plant growth and development than the individual stresses [7]. Due to changing environmental conditions, severity of environmental stresses increases and poses a serious threat to agricultural activities all over the world [8,9].

Wild populations of plants are resistant to different kinds of abiotic stresses including high temperature, extreme drought, and salinity [10]. Structural and functional modifications in plants under specific environmental conditions are evolutionary outcomes that provide the great opportunity to investigate adaptations acquired by plants over time [11,12]. Grasses have different structural modifications such as hairiness on the leaf surface, reduced metaxylem area, stomatal density and area, increased leaf epidermal thickness, dense sclerification, and bulliform cell area [13]. Vascular bundles are surrounded by extensive sclerification, while the size of xylary vessels increase in size [14]. Plants also show physiological modifications to overcome the abiotic stresses, i.e., production of osmolytes such as high sugar content [15] and phenolic compounds [16].

The immediate response to abiotic stresses is the shortage of water, otherwise known as physiological drought [17]. Plants adopt different strategies to overcome the environmental adversaries that may cause water scarcity [18]. Deep root systems are morphological modifications for drought tolerance which enables the plants to conduct water from deeper soil [19]. Stunted growth makes the plants capable of spending energy for survival [20]. Reduction in leaf area helps conserve water by reducing transpiration rate [21]. For salt tolerance, salts excretory structures like salt glands and microhairs are present in grasses growing in saline habitat [22]. Stomatal size and orientation on adaxial or abaxial leaf surfaces are the most important features to control the rate of transpiration [23].

A perennial species, *C. pennisetiformis* (Hochst & Steud) Wipff, commonly called Cloncurry Buffel grass is widely distributed all over the world [24]. It is a highly nutritious grass species used as a fodder for grazing animals [25]. *C. setiger* Vahl (birdwood grass), locally called Anjan ghas, is native to tropical Africa and is widely distributed throughout Southeast Asia and the Middle East [26]. It is a nutritive fodder and forage grass and used as pasture in arid and semi-arid regions [27]. This grass species has excellent capacity to soil-binding to conserve the soil particles in desert regions [28]. Large-spike Buffel grass (*C. prieurii* (Kunth) Maire) is an annual, excellent fodder grass of deserts and semi-deserts of India, Pakistan, Northern Africa, and Middle East. Seeds are edible, often mixed with millets for making bread [29].

The habitats of *C. prieurii* were entirely different from *C. pennisetiformis* and *C. setiger.* The latter species co-existed in a variety of environmental conditions. Previously the individual *Cenchrus* species were evaluated for either salinity or drought stress, but in the present study, their response was evaluated in multiple abiotic stresses for the first time. Moreover, plasticity and variation in structural and functional features in differently adapted populations were not reported in the *Cenchrus* species under observation. The study was conducted to investigate structural and functional modifications of various *Cenchrus* species under abiotic stresses, and to distinguish specific structural responses that are important for water conservation. It was hypothesized that each *Cenchrus* species may adopt different structural and functional strategies to deal with environmental extremities caused by multiple abiotic stresses. The research questions to be addressed included: (1) what are the structural and functional modifications in *Cenchrus* species in response to multiple abiotic stresses; (2) how do these modifications contribute towards successful survival in extreme aridity, salinity or cold; and (3) which structural or functional traits are of ecological significance that contribute towards ecological success of *Cenchrus* species under study.

## 2. Results

### 2.1. Environmental and Soil Physicochemical Traits

Elevation was significantly higher in the cold mountains of all grass species, with the maximum (2446.3 m) being recorded for the *C. setiger* habitat. Elevation of the desert habitats varied between 108.4 to 147.3 m as observed for *C. setiger* and *C. prieurii* habitats respectively (Table 1). Annual rainfall was extremely low in arid regions of all grasses, while exceptionally high in the cold mountainous region. The maximum rainfall was recorded at the cold region of *C. setiger* (1747.2 mm) which receive the maximum snowfall (15.4 mm). The maximum absolute temperature of normal-control, arid, and saline habitats varied between 45 to 50 °C, while the minimum temperature was between 1.1 to 2.1 °C. The cold mountainous habitats were significantly colder, where the maximum absolute temperature ranged between 21 to 24 °C and minimum between −3.1 to −5.6 °C.

In *C. pennisetiformis* habitats, saturation percentage was the highest at normal control (25.6). The arid habitat showed the maximum pH (8.1), K^+^ (112.1), Ca^2+^ and PO_4_^3−^ with the minimum organic matter and saturation percentage. The saline habitats depicted the highest ECe, Na^+^, and NO_3_^−^ while the cold region exhibited the greatest organic matter, saturation percentage, and pH. Normal control habitat of *C. prieurii* exhibited the highest organic matter, pH, and NO_3_^−^, while arid habitat had the maximum soil Ca^2+^. The rhizospheric soil of the saline region showed the maximum ECe, pH, Na^+^, and K^+^, whereas cold mountainous habitats had the highest organic matter, saturation percentage, and PO_4_^3−^. The arid zone habitat of *C. setiger* had the maximum pH, K^+^, and PO_4_^3−^. The greatest saturation percentage, ECe, Na^+^, Ca^2+^, and NO_3_^−^ were observed in rhizospheric soil of the saline habitat, while organic matter and NO_3_^−^ in cold mountainous region, respectively (Table 1).

### 2.2. Proportion of Morpho-Physiological Traits

The raw data is presented in Appendix A. *C. pennisetiformis* showed the maximum percentage of root length (51.3%), number of leaves per plant (38.3%), shoot fresh (30.8%), and dry weights (33.1%) in the arid conditions (Figure 1). Inflorescence length increased incredibly (59.0%) in the cold mountainous region in this species. The percentage of leaf area (48.0%), root fresh (38.5%), and dry weights 37.6%) were the maximum in normal control habitats. In *C. prieurii*, the maximum root length (56.2%), leaves per plant (36.4%), and leaf area (61.8%) percentages were observed in saline conditions. Root fresh (49.9%) and dry weights (52.6%) were the highest in *C. prieurii* in plants from cold mountains. There was little variation regarding morphological traits in *C. setiger* collected from different habitats where only the percentage of leaf area (41.2%) increased significantly in the arid region (Figure 1).

### 2.3. Proportion of Physiological Traits

*C. pennisetiformis* showed the greatest percentage of total soluble proteins (35.7%), amino acids (51.1%), and proline (37.9%) under saline conditions, while in *C. prieurii* the highest percentage of total soluble proteins (30.7%), amino acids (48.2%), and proline (30.6%) was in plants from arid conditions (Figure 2). The percentage of glycine betaine (35.4%) was the highest in *C. prieurii* from the normal control habitat. In *C. setiger*, the percentage of total soluble proteins (37.8%) was the maximum in plants from cold mountains.

Percentage of chlorophyll *a* (35.8%), *b* (33.7%), and total chlorophyll (35.0%) were the maximum in *C. pennisetiformis* collected from arid conditions (Figure 2). Chlorophyll *a* (27.0%) and carotenoids (56.8%) percentages were the greatest in *C. prieurii* from cold mountains. Carotenoids percentage was the maximum (35.8%) in *C. pennisetiformis* from saline conditions. Percentage of total chlorophyll was the highest (36.1%) in *C. prieurii* from the normal control habitat. Shoot Na^+^ (40.4%) and K^+^ (30.9%) percentages were the highest in *C. prieurii* collected from saline habitats. Shoot K^+^ percentage was the highest (30.9%) in *C. prieurii* collected from saline area and the highest Ca^2+^ (30.8%) was from arid regions.

### 2.4. Proportion of Dermal Tissue

Root epidermis was not recorded in *C. pennisetiformis* under saline and cold conditions because soil friction removed the delicate epidermis in plants colonizing these sites (Figure 3, Figure 4, Figure 5, Figure 6 and Figure 7). The maximum epidermal thickness percentage (52.5%) was noted in *C. pennisetiformis* plants from arid regions. In *C. prieurii*, percentage of root pericycle thickness (45.1%) was the highest in the arid condition. The proportion of stem epidermal thickness (44.1%) was the maximum in *C. setiger* in plants from the cold mountainous region. The percentage of leaf epidermal thickness (37.2%) was the highest in *C. pennisetiformis* collected from arid conditions. In *C. prieurii*, the proportion of leaf bulliform thickness (44.1%) was the highest in plants collected from arid conditions, whereas in *C. setiger* the thickest bulliform cells (47.8%) were noted in plants from the normal control habitat. Adaxial stomatal area percentage was maximum in *C. pennisetiformis* (32.2%) and *C. setiger* (33.1%) under the normal control habitat. The proportion of abaxial stomatal area (29.5%) was the highest in *C. setiger* plant collected from the normal control habitat.

### 2.5. Proportion of Mechanical Tissue

The proportion of stem sclerenchyma thickness was the maximum in *C. pennisetiformis* (31.5%) and *C. setiger* (36.4%) plants from cold mountainous regions, while in *C. prieurii* the highest percentage of sclerenchyma thickness (37.7%) was noted in plants from saline areas. The percentage of leaf sheath sclerenchyma thickness was the maximum in *C. pennisetiformis* (32.0%) and in *C. prieurii* (30.0%) collected from cold mountainous regions.

### 2.6. Proportion of Areas and Thickness Root, Stem and Leaves

Root radius percentage was the highest (40.4%) in *C. pennisetiformis* from the normal control habitat, while it was the maximum (33.0%) in *C. prieurii* roots from arid conditions. In *C. setiger*, percentage of root radius was the maximum (31.4%) in plants collected from cold mountainous region (Figure 3, Figure 4, Figure 5, Figure 6 and Figure 7). Stem radius showed little variation in *Cenchrus* spp. growing in different environmental conditions. The proportion of leaf sheath thickness was the highest in *C. pennisetiformis* (46.9%) and *C. setiger* (37.6%) plants collected from arid conditions. In *C. prieurii*, plants colonizing cold mountains exhibited the highest proportion of leaf sheath thickness (39.2%). The leaf midrib proportion was the highest in *C. pennisetiformis* (37.0%) and *C. prieurii* (38.3%) plants from arid environments, while in *C. setiger*, it was the greatest (44.1%) in normal control habitat. The percentage of lamina thickness was the highest (31.9%) in *C. prieurii* plants from drought conditions, while there was little variation in other species.

### 2.7. Proportion of Parenchymatous Tissue

The percentage root cortical thickness was the greatest in *C. pennisetiformis* (44.9%) and *C. prieurii* (37.3%) plants from arid zones, while in *C. setiger* it was the maximum (39.3%) in plants from cold mountains. The proportion of root aerenchyma was the highest in *C. pennisetiformis* (53.3%) and *C. prieurii* (68.5%) from the normal control habitat (Figure 8). The root aerenchyma percentage (Figure 4, Figure 5, Figure 6 and Figure 7) was the highest (50.0%) in *C. setiger* collected from saline conditions. Root pith thickness percentage was the greatest in *C. pennisetiformis* (37.3%) plants from normal control habitat. In *C. prieurii*, the root thickness proportion was the maximum (34.2%) in the arid environment. The pith thickness percentage was the maximum (41.8%) in *C. setiger* collected from cold mountainous regions. The stem cortical thickness proportion was the highest (39.1%) in *C. pennisetiformis* collected from normal control habitat, while in *C. prieurii* (39.4%) and *C. setiger* (48.7%), the maximum stem cortical thickness was observed in plants from cold habitats. The percentage of leaf sheath parenchymatous cell area was the highest in all *Cenchrus* spp. collected from arid conditions, measuring as 80.4% in *C. pennisetiformis*, 47.9% in *C. prieurii*, and 35.9% in *C. setiger*. The proportion of leaf parenchymatous thickness was the highest in *C. pennisetiformis* (63.5%) and *C. prieurii* (39.5%) plants collected from saline conditions. In *C. setiger* leaf parenchymatous thickness percentage was the highest (58.5%) in normal control habitat. Leaf mesophyll thickness varied slightly in all *Cenchrus* species.

### 2.8. Proportion of Vascular Tissue

Root metaxylem area ratio in *C. pennisetiformis* was the maximum (38.3%) in plants collected from arid conditions. The root phloem area proportion was the highest in *C. prieurii* (38.5%) and *C. setiger* (40.5%) plants collected from arid environments. The stem metaxylem ratio was the highest (35.1%) in *C. pennisetiformis* collected from cold mountains. Stem metaxylem fraction was the highest (43.2%) in *C. setiger* collected from arid environments. In *C. setiger* stem vascular bundle ratio was the greatest (37.9%) in normal control habitats. In *C. prieurii*, stem phloem proportion was the highest (36.2%) in plants collected from arid conditions. The stem phloem area percentage in *C. setiger* was the maximum (53.0%) in plants collected from normal control. In *C. prieurii*, leaf sheath vascular bundle area was the highest in cold, mountainous plants. In *C. setiger*, the proportion of leaf sheath vascular bundle area was highest (44.3%) in population from arid habitats (Figure 8). The maximum leaf metaxylem area ration was the highest (54.5%) in *C. setiger* plants collected from normal control habitat. The leaf phloem area percentage was the highest (36.2%) in *C. pennisetiformis* from saline conditions. In *C. prieurii*, a fraction of leaf phloem area was the greatest in saline and cold mountainous conditions. The leaf phloem area ratio was the maximum in *C. setiger* collected from normal control (36.7%). The leaf vascular bundle area proportion was the maximum in *C. pennisetiformis* from drought (35.2%) and saline (37.1%) habitats, while in *C. prieurii*, it was the highest (33.5%) in arid conditions.

### 2.9. Pearson’s Correlation Coefficients (p < 0.05) for Environmental/Soil and Morpho-Anatomical and Physiological Traits

Morphological traits such as plant height, root length, and shoot fresh weight were positively correlated with soil organic matter (Table 2). The number of leaves per plant was positively correlated with soil Na^+^ but negatively with soil NO_3_^−^. Inflorescence length was positively linked with elevation. Root cortical thickness was positively associated with organic matter, while root phloem area was positively correlated with soil Ca^2+^. Root pith thickness was negatively correlated with soil Na^+^. The metaxylem area was positively correlated with soil ECe and pH. Stem vascular bundle was negatively associated with ECe. The stem phloem area was negatively correlated with soil Na^+^ but positively correlated with soil Ca^2+^. Leaf sheath epidermis thickness was positively correlated with soil organic matter. Vascular bundle area was positively correlated with elevation. Leaf epidermal thickness and phloem area were positively correlated with soil Ca^2+^. The parenchymatous cell area was positively correlated with saturation percentage. Midrib thickness was positively correlated with soil K^+^, while lamina thickness negatively correlated with soil Na^+^. Adaxial stomatal density was negatively associated with soil K^+^. Adaxial and abaxial stomatal area were positively correlated with soil Ca^2+^. Total soluble proteins were negatively associated with soil K^+^, while glycine betaine positively correlated with soil organic matter. Chlorophyll *a* and *b* were positively associated with soil Ca^2+^. Shoot Na^+^ was negatively correlated with elevation and soil saturation percentage, but positively associated with soil ECe. Shoot Ca^2+^ was negatively correlated with saturation percentage while positively correlated with soil ECe (Table 2).

### 2.10. Principal Component Analysis Showing Relationship among Soil/Environmental and Plant Traits

Principal component analysis (PCA) biplot among soil/environmental, morphological, and physiological attributes showed three isolated clusters for each *Cenchrus* species (Figure 9). In the first cluster, plant height and leaf area of *C. pennisetiformis* was strongly associated with soil K^+^, Ca^2+^ and PO_4_^3−^ in cool mountainous habitats. In the second cluster, *C. setiger* showed a strong relationship with shoot K^+^, shoot dry weight, and number of leaves per plant under saline conditions. *C. setiger* was closely associated with shoot dry weight and organic matter from arid conditions. Under cool conditions, *C. setiger* was strongly associated with soil Na^+^ and soil pH. The third clusters showed a strong association between glycinebetaine, root length, and soil ECe in *C. prieurii* plants from saline conditions. Under drought conditions, *C. prieurii* was strongly associated with shoot Ca^2+^, total free amino acids, total soluble proteins, chlorophyll *b*, and proline.

*C. prieurii* was associated with soil NO_3_^−^ and total chlorophyll under the normal control habitat. *C. setiger* exhibited association of stem metaxylem, soil ECe, and Na^+^ in cold and arid conditions. *C. pennisetiformis* showed a close relationship of root pericycle thickness, cortical cell area, stem vascular bundle area, stem epidermal thickness with soil NO_3_^−^, saturation percentage, moisture content, and elevation under saline conditions. Under arid conditions, root aerenchyma, cortical region thickness, root metaxylem, and phloem area of *C. pennisetiformis* was strongly associated with soil organic matter. Under normal control habitat, *C. pennisetiformis* showed a close association with root radius. *C. prieurii* showed strong association of root phloem area and endodermal thickness with soil Ca^2+^ under saline and drought conditions. Under a normal control habitat, *C. prieurii* showed a close relationship of soil K^+^, annual minimum temperature, with root traits such as pericycle thickness, pith area, epidermis thickness, and stem traits such as phloem area, sclerenchymatous thickness, and stem radius (Figure 9).

*C. setiger* showed close relationship of soil Na^+^, ECe and pH with cold and saline conditions. *C. prieurii* showed a strong association of soil NO_3_^−^ with leaf epidermal thickness, midrib thickness, lamina thickness, and vascular bundle area under cold and arid environments. Under saline conditions, *C. prieurii* annual minimum temperature, soil Ca^2+^, was strongly associated with abaxial stomatal area and mesophyll thickness. *C. prieurii* under normal control habitat showed close relationship with parenchymatous cell area, saturation percentage, soil K^+^, and adaxial stomatal area. The soil organic matter and PO_4_^3−^ in rhizosphere of *C. pennisetiformis* under the normal control habitat was closely associated with leaf sheath thickness and adaxial stomatal density. Under arid and saline conditions, the elevation of *C. pennisetiformis* showed a strong relationship with parenchymatous cell area, epidermal thickness, bulliform area, sclerenchymatous thickness, and abaxial stomatal density (Figure 9).

### 2.11. Relationship between Soil/Environmental and Morpho-Physiological Traits Anatomical Traits

The heatmap among soil/environment, morphological, and physiological traits is presented in Figure 10. Grass species under different habitats showed three isolated clusters. In the first group, *C. setiger* under drought and cold, *C. pennisetiformis* under drought and *C. prieurii* under salinity were closely clustered. In the second group. *C. pennisetiformis* under normal control and cold, and *C. setiger* under normal control and salinity were clustered together. The third group showed association of *C. pennisetiformis* under salinity and *C. prieurii* under normal control, drought and cold were closely clustered. *C. setiger* under drought showed strong positive correlation with soil ECe and negative correlation with saturation paste. *C. setiger* under cold was negatively correlated with minimum temperature. *C. prieurii* under salinity was positively associated with root length and negatively with soil NO_3_^−^. *C. pennisetiformis* under cold was positively correlated with inflorescence length, while negatively correlated with chlorophyll *a*, *b*, minimum temperature, and soluble sugars. This species under normal control was positively related to plant height and negatively associated with soluble sugars, soluble proteins, and proline. A negative association was recorded in *C. setiger* with soluble proteins, while those collected from the saline region showed positive correlation with soil Na^+^ and negative correlation with soil organic matter and PO_4_^3−^. *C. prieurii* collected from cold mountains exhibited positive association with root fresh and dry weights and negative association with soil organic matter, minimum temperature, and shoot Na^+^. *C. pennisetiformis* under saline was negatively correlated with shoot fresh and dry weights and soil Ca^2+^. *C. prieurii* under normal control possessed a positive relationship with soil total chlorophyll and negatively associated with PO_4_^3−^, whereas under drought a negative association was recorded with soil saturation percentage and moisture content.

### 2.12. Relationship between Soil/Environmental and Root/Stem Anatomical Traits

Three distinct clusters were observed in a heatmap among soil/environment and root/stem anatomical traits. *C. prieurii* under drought and salinity and *C. pennisetiformis* under normal control and drought were clustered in close association. *C. pennisetiformis* and *C. setiger* under cold were clustered together. A large cluster was observed for *C. prieurii* under cold and normal control, *C. setiger* under salinity and normal control, and *C. pennisetiformis* under salinity while *C. setiger* under drought responded independently. *C. setiger* under drought showed a positive correlation with soil pH, ECe, and maximum temperature, and a negative correlation with stem vascular bundle sclerenchymatous thickness. *C. prieurii* under aridity was positively correlated with stem phloem, root metaxylem, and soil Ca^2+^, whereas under salinity it was positively associated with root metaxylem. A positive relationship was recorded for *C. pennisetiformis* under normal control with root radius, root aerenchyma area, and stem cortical thickness, whereas under drought this species showed positive association with root cortical thickness. In *C. pennisetiformis* under cold, a positive association was observed with elevation, while negative with root epidermis. In *C. setiger* under cold, a positive correlation was observed with stem epidermis, and a negative correlation was observed with minimum temperature. *C. setiger* under salinity was positively correlated with soil Na^+^ and negatively correlated with root pith area. *C. setiger* under normal control habitat showed a positive correlation with soil K^+^ (Figure 10).

### 2.13. Relationship between Soil/Environmental and Leaf Sheath/Leaf Anatomical Traits

Clustered heatmap among environmental/soil and leaf sheath/leaf anatomical traits showed four distinct clusters. In the first group, *C. pennisetiformis* under arid, saline, and cold conditions were clustered in a close association. In the second group, *C. setiger* under arid and cold conditions were clustered together. In the third group, saline and normal control populations of *C. setiger* and *C. pennisetiformis* were clustered in close association. The fourth group has cold, normal control, arid, and saline populations of *C. prieurii* closely clustered. *C. pennisetiformis* showed a positive correlation with parenchymatous cell area and epidermal thickness, whereas a negative correlation was noted with pH under arid conditions. Under saline conditions, *C. pennisetiformis* was positively correlated with adaxial stomatal density and abaxial stomatal density and negatively correlated with soil Ca^2+^. *C. pennisetiformis* under cold conditions was positively associated with elevation. *C. setiger* under arid conditions showed positive correlation with ECe, while a negative correlation with lamina thickness. Under cold, *C. setiger* showed negative correlation with minimum annual temperature. *C. setiger* showed a positive association with soil Na^+^ under saline conditions. *C. setiger* was negatively correlated with abaxial stomatal density in the normal control habitat. *C. prieurii* under cold conditions showed a positive association with vascular bundle area and a negative correlation with organic matter. Under drought conditions, *C. prieurii* showed a negative relationship with saturation percentage and moisture content (Figure 10).

### 2.14. Estimated Response of Structural and Functional Traits

Estimated responses of different traits, i.e., morphology, root anatomy, stem anatomy, leaf sheath anatomy, leaf anatomy, and physiological traits among three *Cenchrus* species are presented in Figure 11. Among morphological features, a positive response of *C. setigerus* and *C. pennisetiformis* was recorded, while *C. setiger* responded negatively (Figure 11a). The root anatomical traits showed strong deviation, whereas only a few traits of *C. pennisetiformis* responded positively (Figure 11b). Stem anatomical traits showed a linear response, where *C. pennisetiformis* showed a positive association to climatic factors (Figure 11c). In leaf heath traits a linear trend was noted where *C. pennisetiformis* and *C. prieurii* responded positively (Figure 11d). Leaf anatomical traits showed non-linear trend and strong deviation, where *C. pennisetiformis* and *C. prieurii* responded positively (Figure 11e). Physiological traits showed a parallel trend with least deviation, where a few traits responded positively while others responded negatively (Figure 11f).

## 3. Discussion

The *Cenchrus* species (*C. pennisetiformis C. prieurii*, and *C. setiger*) are widely distributed and can colonize different habitats through specific structural and functional alterations [30]. *C. pennisetiformis* and *C. setiger* are widespread grass species that can tolerate environmental stresses such as drought, cold and hot temperatures, waterlogging, and salinity [31]. These two species remain green under hyper-arid conditions and usually co-exist in a variety of habitats [32]. *C. prieurii* is a typical desert species that can tolerate drought, salinity, and cold conditions [32]. Selected grasses were collected from different stressful environments like normal (control), drought, salinity, and cold mountainous region.

### 3.1. Cenchrus pennisetiformis

*C. pennisetiformis* is a perennial grass and grows in different habitats like moist shady places, snowy areas, and saline patches, and can survive under harsh conditions [30]. Arid regions suited the development of morphological traits such as root length, number of leaves per plant, and root and shoot fresh and dry weights, which were the highest population from arid region. Inflorescence length was more developed in plants of cold mountainous regions where the plants height was greatly reduced. Cold stress enhances plant growth and biomass production, which may result in a bushy appearance [33]. This species is relatively less tolerant to salinity stress [34]. The population collected from the saline area showed minimum growth and development of underground and aboveground plant organs. The proportion of physiological attributes like total soluble proteins, soluble sugars, glycine betaine, chlorophyll *a*, *b*, and total chlorophyll increased in drought affected population. Better accumulations of organic osmolytes and increased chlorophyll content indicate better adaptability of this species to arid environments [35,36]. The total proteins, total free amino acids, proline content and carotenoid significantly increased in cold habitat plants. This confers high tolerance to cold stress as increased accumulation of organic osmolytes is critical for cell turgor maintenance [37,38].

The proportion of dermal tissues in roots, leaf sheath, and leaf blade increased in the population exposed to drought stress. The root epidermis was absent in saline and cold affected populations which was likely due to soil friction. Dermal tissue acts like a barrier to movement of water outwards [39]. Stem mechanical tissue was greatly developed due to lignin deposition in population facing cold stress. Mechanical tissue prevents tissue collapse and reduces water loss from leaf sheath surface [40]. Intensive sclerification outside the stem vascular region is extremely beneficial for plants facing hyper arid conditions [41].

The proportion of parenchymatous tissues in roots leaf sheaths was the greatest in drought-affected population. This trait was greatly reduced under salinity stress. Leaf sheath parenchymatous tissue proportion was also greatly reduced under cold stress. Drought stress causes water scarcity in plants which generally hampers growth and development. In species like *C. pennisetiformis*, drought improved growth and development of parenchymatous tissue. Ref. [42] reported anatomical alternations in plants. Parenchymatous tissue enhances the water storage capacity and is extremely important in water deficit conditions [26]. Leaf parenchymatous thickness was the highest in the population exposed to cold stress. Parenchyma tissues were more developed under cold stress and helpful for water conservation, which is critical for the survival of this species under environmental adversaries [30]. Modifications in dermal and parenchymatous tissues ensure ecological success of *C. pennisetiformis* in a variety of habitats [26].

The proportion of vascular tissue in roots and leaves increased under drought stress. Root length, cortical thickness, phloem area, vascular bundle area, and leaf midrib thickness are key anatomical features under harsh environmental conditions [43]. Vascular tissue in stem increased under cold stress, while root vascular tissue decreased under cold affected population. Increased size of phloem and metaxylem vessels are related to better water and solute conduction under stressful conditions [44]. Leaf vascular tissue increased in the saline affected population.

### 3.2. Cenchrus prieurii

*C. prieurii* populations were collected from various habitats such as cool mountains, along roadsides, hyper-arid and semiarid regions, and along riverbanks. Morphological traits such as root length, leaves per plant and leaf area were the greatest in populations from saline areas, whereas root fresh and dry weights were in population from cold mountains. Longer roots can extract water from deeper soil layers in addition to providing structural strength of plants under stress conditions [45]. The increased biomass production under cold in this species is the good criteria for estimating stress tolerance [46].

Physiological traits like photosynthetic pigments, organic osmolytes (total free amino acids) and ionic content were the greatest under drought stress, while carotenoids increased under cold stress. Physiological parameters play an important role, such as in the anatomical features for the existence of the species under water deficit conditions caused by environments stresses. High concentration of ionic content, especially K^+^ and Ca^2+^, are helpful for the better growth and biomass production of a species under abiotic stresses [47].

Areas and thickness like root radius, midrib thickness, and leaf lamina thickness increased in drought-affected populations. Leaf sheath thickness was greatly reduced in population from saline area, while this trait greatly increased the cold region population. The leaf sheath thickness was probably due to the high proportion of parenchymatous cells. Any increase in this tissue will increase in water storage capability, hence enabling the plants to survive under prolonged period of harsh environment [48].

The proportion of dermal tissues was higher in population of *C. prieurii* from drought-affected area, while root epidermal thickness decreased under salinity stress. The most important cells in epidermis of grasses are the bulliform cells, which control leaf rolling. These cells are deeply inserted in leaf epidermis and protect adaxial epidermis and stomata by leaf rolling, hence immensely important for water conservation [49]. Sclerification due to lignin deposition in stem is the most critical alteration in response to drought or physiological drought. Under saline stress, increased sclerification in stem plays a significant role for water conservation as it prevents outward movement of water in addition to providing mechanical strength to metabolically active tissues [43].

Parenchyma tissues in root and leaf sheath increased under drought stress. The increased storage parenchyma in *C. prieurii* is vital for water storage, and this confers better adaptation of *C. prieurii* for hyper-arid environments. Root aerenchyma was absent in drought- and salt-affected populations. Aerenchyma formation is a characteristic of aquatic plants [50] but has also been reported in grasses growing under saline or arid conditions [22,30]. Among vascular tissue, root metaxylem area was the highest in saline population. Leaf sheath vascular bundle area increased in the population facing cold stress. Increased vascular bundle area in population from cold mountains indicates a high degree of stress tolerance [48].

### 3.3. Cenchrus setiger

*C. setiger* is adapted to a variety of abiotic stresses such as drought, salinity, and cold climatic conditions [51]. Increase in root length in the *C. setiger* population colonizing arid region is among the major factors for survival in water deficit conditions [Alvarez 8]. Increased leaf area in population from arid regions is a good indicator of high degree of drought tolerance in *C. setiger* as it maintains growth and development in stressful conditions [32].

Physiological traits such as organic osmolytes and total soluble proteins in *C. setiger* increased in populations collected from the drought and cold, while proline was increased in drought affected population. High concentration of osmolytes is important for the maintenance of turgor pressure under hyper-arid and extreme cold circumstances [52]. Photosynthetic pigment like carotenoid was increased in population from cold stress but decreased in saline stress. High concentration of chlorophyll pigments has earlier been reported by Fatima et al. [53] in *Dichanthium annulatum* from high elevations.

The proportion of dermal tissue such as root epidermis increased under drought affected population and decreased in populations exposed to salinity and cold stresses. Root endodermal and stem epidermal thickness significantly increased under cold stress, while leaf bulliform thickness was the maximum in normal control population. This may be due to enough water and bulliform cells acting as a storage parenchyma [54]. Endodermal thickness in roots [55] and well developed bulliform cells [56] is an indicator of a high degree of aridity tolerance of grasses.

In mechanical tissue, stem sclerenchyma thickness increased under cold stress. In plants inhabiting high elevation, sclerification outside the stem vascular region not only provides the structural and mechanical support of plant species but also reduces the water loss [13]. Root radius and leaf sheath thickness increased in cold-affected population. Leaf sheath thickness and root radius is due to high percentage of storage parenchyma (or stelar region), which play an important role in water storage and critical under deficit condition Wu et al. [57]. Midrib thickness increased under salinity stress, and was greatly reduced under drought and cold stresses. Similar findings have been reported by Fatima et al. [13] and Kasirajan et al. [58] in salt tolerant/halophytic species. Anatomical modifications such as midrib and lamina thicknesses are related to succulence [13].

The proportion of parenchymatous tissue of *C. setiger* in root and stem such as cortical thickness and pith thickness greatly increased under cold stress. A high proportion of parenchyma in population facing cold stress is the indication of high degree of stress tolerance [59]. The leaf parenchyma tissue was the highest in normal control population, but greatly reduced in drought affected population. A decreased proportion of parenchyma resulted in a reduction of leaf thickness, hence making the leaves much easier to roll and protecting the adaxial surface from direct contact to external environment [22]. This modification reduces transpiration rate significantly and is critically beneficial for survival under hyper-arid climates [60]. Leaf thickness, in contrast, is an important criterion for ultimate degree of stress tolerance as thicker leaves can persist for longer periods under water scarce conditions [40]. Pith thickness is greatly reduced under salinity stress indicating its low tolerance to salinity stress Wasim et al. [43]. Root aerenchyma cavities increased in population collected from saline area. Aerenchyma is the most prominent feature of plants growing in saltmarshes, that is responsible for the gases exchange efficiently throughout the plant [61].

Vascular tissue was generally more developed in population collected from hyper-arid habitats. The proportion of root metaxylem area and phloem area, stem metaxylem area, and leaf sheath vascular bundle area was the maximum. Broad metaxylem in herbaceous grasses like *C. setiger* is extremely beneficial under aridity [22] as it facilitates the conduction of solutes [53]. Large vascular bundles supported by mechanical tissue especially on both sides are characteristics of desert grasses [49]. This protects inner tissues from desiccation and is the major factor responsible for the survival of *C. setiger* under multiple environmental stresses [62].

Overall response of *Cenchrus* species to environmental stresses is presented in Table 3. All *Cenchrus* species responded differently, not only in the normal control environment but also under stressful conditions. *C. pennisetiformis* population were the tallest with the largest leaf area, but the drought-affected population showed longest root, produced more leaves, and shoot fresh and dry biomass. This is an indication of bushy appearance, hence the perfect adaptation for hyper-arid habitats. Total soluble sugars were among the osmolytes that accumulated in the highest concentration in this population. Moreover, root and leaf epidermis, root, leaf sheath parenchymatous tissue, and root metaxylem area were the maximum. These modifications are critical for minimizing water loss from plant surface, storing additional water in leaf sheath and efficient solute conduction through xylem vessels. Salinity inhibited growth and development in *C. pennisetiformis*. Total soluble sugars, chlorophyll *a,* and total chlorophyll increase markedly under salinity, as were the leaf parenchyma and phloem proportion. Root cortical thickness and metaxylem vessel area greatly reduced, indicating sensitivity of this ecotype to salinity stress. Under cold stress, the most remarkable feature was increased inflorescence length. Shoot Na^+^ and sclerification in stem increased significantly, while root phloem decreased. Increased lignification is one of the major factors for survival in extreme cold environments.

*C. prieurii* showed better shoot fresh and dry weights in ecotypes collected from normal control, root length, leaf number, and area in the salinity ecotype and root fresh and dry weights in the ecotype facing cold stress. This indicated better tolerance of *C. prieurii* ecotypes growing in saline and cold environments. Glycine betaine accumulated in the ecotype from normal control, total free amino acids, and shoot Na^+^ in the ecotype from drought area and carotenoids from cold mountains. At stem level, stem phloem increased in the ecotype from hyper-arid habitat, sclerification in the ecotype from saline area and cortical parenchyma in the ecotype exposed to cold stress. Several leaf anatomical traits such as bulliform thickness, leaf (midrib and lamina) thickness metaxylem area, and vascular bundle area increased in drought prone habitats, indicating better tolerance of this species to hyper-arid environments. Root traits such as root radius, pericycle thickness and pith area increased in the ecotypes from arid environments.

In *C. setiger*, leaf anatomical traits such as midrib thickness, parenchyma area, mesophyll area, metaxylem area, vascular bundle area and bulliform thickness were more developed in the ecotype collected from normal control. The ecotype facing drought showed better development of leaf sheath anatomical traits such as thickness, parenchyma area and vascular bundle area. Root traits such as endodermal thickness, cortical region thickness and pith area were more developed in the ecotype exposed to cold stress.

## 4. Materials and Methods

Three *Cenchrus* species (*C. pennisetiformis*, *C. prieurii*, and *C. setiger*) were collected from different ecological regions in the Punjab and Khyber Pakhtoonkhwa during July 2020 to September 2020 (Figure 12). The natural populations of these species were evaluated for their structural and functional responses to multiple abiotic stresses (drought, salinity and cold). The arid habitats were collected from sandy deserts that receive 150 mm or lower rainfall annually. The saline habitats were selected based on soil salinity with ECe more than 6 dS m^−1^. Populations exposed to cold stress were collected from the habitats where minimum temperature falls below 0 °C and receive above 300 mm snowfall during winters. Annual maximum temperature of the normal-control habitats ranged from 43 to 45 °C and salinity below 3 dS m^−1^. *C. pennisetiformis* collection sites were Manara, Salt Range (normal control), Angoora Farms, Thal Desert (drought), Khewra foothills, Salt Range (salinity), and Bansra Galli, Murree (cold). *C. prieurii* habitats were Gatwala, Faisalabad (normal control), Nawan Kot, Thal Desert (drought), Musa Khal, Mianwali (salinity) and Nathia Galli (cold), while *C. setiger* habitats were Pindi Battian, Hafizabad (normal control), Nawab Din Fort, Cholistan Desert (drought), Sahianwala, Faisalabad (saline) and Thandiani, Abbottabad (cold).

### 4.1. Sample Collection

The plant samples were collected carefully with soil auger (20 cm dia.) and preserved in plastic zipper bags. The material was then kept in an icebox for further physiological and anatomical analysis in the laboratory. Sampling layout is presented in Figure 13. Ten quadrats (1 m × 1 m) were placed along a transect line (100 m) at each study site keeping in view where the pure community of each species existed. This made the maximum probability (about 95%) of the existence of a plant in each quadrat as mentioned in Figure 3. If a plant was absent from a particular quadrat then the adjacent quadrat was selected. Three quadrats were selected at each study site for morphological, anatomical, and physiological studies (indicated by red colour in Figure 13). One plant within each red quadrat was selected. Three transect lines (replications) were positioned at each habitat, each separated by 250 m. Data of three plants were then averaged for each replication, and then used in the analyses of different traits (Figure 13). The vegetation analysis was conducted from all 10 quadrats laid at each transect line, which is not presented in the present study.

### 4.2. Physiographic Data

Geographic data like coordinates and elevation of each collection site were recorded by Global Positioning System (Garmin, eTrex Venture HC, Garching bei München, Germany). Meteorological data was collected from the Meteorological Department of Pakistan Islamabad (https://rmcpunjab.pmd.gov.pk/metData.php, accessed on 23 November 2002).

### 4.3. Soil Analysis

Soil was taken near the rhizosphere of each population at a depth of 15–20 cm to analyze the physicochemical traits (Table 1). The soil sample (200 g) was taken, mixed thoroughly, and then completely dried in an oven at 70 °C for one week. Saturation paste was prepared for the estimation ECe, saturation percentage and ionic. Saturation percentage content as measured by a formula:Saturation percentage = Weight of a saturated paste − Dry weight of soil

Soil ECe and pH were calculated by portable pH/Electrical Conductivity Meter (WTW series InoLab pH/Cond 720, Xylem, Washington, DC, USA) following the methods labeled in Handbook No. 60 [63].

Soil samples were extracted with deionized water. Soil samples were digested in mixture of nitric acid/perchloric acid (3:1 ratio) for the estimation of Na^+^ and K^+^ with flame-photometer (Model 410, Sherwood Scientific Ltd., Cambridge, UK). The Ca^2+^ and Mg^2+^ were recorded with atomic absorption spectrophotometer (AAnalyst 3000; Perkin Elmer, Rodgau, Germany). Soil NO_3_^−^ and PO_4_^3−^ were estimated spectrophotometrically with the methods of Kowalenco [64] and Yoshida [65], respectively. Soil Cl^−^ was recorded with chloride meter (Model 926; Sherwood Scientific Ltd., Cambridge, UK).

### 4.4. Morphological Traits

Data was taken after the collection of plant samples (randomly collected three samples from each population of each species). Plant height was recorded from the base of the stem to the top. For the leaf area, five leaves at fixed locations of each plant were measured, Leaf area was calculated according to Schrader [66] by the formula:Area = Length × width × correction factor (0.71)

The average leaf area was calculated by each leaf multiplied with total number of leaves per plant. Plant fresh weight was recorded by portable digital balance immediately after uprooting of plant in the field. Dry weight was measured after completely drying of plant material in an oven for one week at 60 °C. Total inflorescence length of each plant was measured by a scale.

### 4.5. Physiological Traits

#### 4.5.1. Ionic Content

The ionic content of root and shoot were measured by using standard method of Wolf [67] using flame photometer (PFP-7, Jenway, UK).

#### 4.5.2. Photosynthetic Pigments

Photosynthetic pigments such as chlorophyll *a* and *b* were measured according to the standard procedures of Arnon [68] and carotenoids were estimated by Scott [69] method using a spectrophotometer (IRMECO, U2020, Lütjensee, Germany).

#### 4.5.3. Biochemical Traits

Proline was measured using spectrophotometer (IRMECO U2020, Lütjensee, Germany) following the method of Bates [70]. For the determination of glycine betaine, a UV-visible spectrophotometer (Hitachi-220, Tokyo, Japan) was used following the procedure of Grattan et al. [71]. Bradford [72] protocol was followed to determine total soluble proteins. Total soluble sugars were determined as reported by DuBois [73]. Total free amino acids were measured according to the protocol proposed by Hamilton [74].

### 4.6. Anatomical Traits

Fresh plant material was collected from the field and immediately preserved in a leak-proof plastic bottle containing formalin acetic alcohol. One-liter solution was prepared as follows:Formalin 50 mL + Acetic acid 100 mL + Ethanol 500 mL + Distilled water 350 mL

The material was subsequently transferred to acetic alcohol solution after 48 h, one liter solution was prepared as:Acetic acid 250 mL + Ethanol 750 mL

Permanent slides of transverse sections were prepared by free-hand sectioning. The sections were dehydrated by serial grades of ethyl alcohol following Ruzin [75]. The sections were stained with a standard double staining technique. Safranin was used for staining tissues with secondary walls (sclerenchyma, xylem vessels), whereas fast green was used for primary walls, (phloem sieves and parenchyma). Canada balsam was used as a mounted material. Photographs of the sections were taken by a camera-fitted digital compound microscope (Meiji Techno Japan, Saitama, Japan). The data were taken by ocular micrometer. Cell area was calculated in accordance with Naz et al. [76]:Area = (Maximum length × Maximum width)/2 × π

### 4.7. Statistical Analysis

The data were converted to percentages of different traits calculated as:Percentage = (Stress (drought, salinity or cold) − Normal control)/(Normal control)

The data were analyzed by analysis of variance in completely randomized design with three replications using statistical software Costat Ver. 6.303. The significance level of means was compared by Duncan’s multiple range test and the least significant difference of each trait was calculated following Steel et al. [77]. Correlation was calculated by statistical software XLSTAT (V. 2021.1).

Relationships between soil/environmental factors and plant structural and functional traits were analyzed by principal component analysis (PCA) using R Studios (V 1.1.463). Heatmaps were constructed between soil/environmental traits and plants traits using a customized R code. A GLM (Generalized Linear Model) was fitted to estimate response of excreted ions and different morpho-anatomical and physiological traits under various abiotic stresses.

## 5. Conclusions

Structural and functional alterations in all *Cenchrus* species were very specific, which is critical for survival under different environmental stresses like hyper-aridity, salinity, and cold. The ecological fitness of these species depended on maintenance of growth and biomass production under stressful environmental conditions.

Plants of *C. pennisetiformis* under arid conditions were shorter in height but heavier in shoot fresh and dry weights, i.e., attained more bushy habit that is a characteristics of desert plants. Survival of this species in arid environments relied on accumulation of compatible solutes and thicker epidermis, which maintained cell turgor and minimized water loss from the plant body. Under salinity as was observed in the arid environments, this species accumulated organic osmolytes and increased proportion of storage parenchyma, critical for water conservation. Under cold environments, *C. pennisetiformis* increased mechanical tissue in the stem which prevented softer tissue from collapsing. Plant height was not affected in *C. prieurii* under abiotic stresses, while shoot biomass was higher under arid conditions and root biomass in cold environments. Leaf anatomical traits such as leaf thickness, dermal tissue, and vascular tissue were more developed in arid conditions indicating aridity as the most suitable habitat for this species. Normal-control habitats were ideal for the growth and development of *C. setiger* regarding leaf anatomical traits. The structural and functional modifications in *Cenchrus* species, such as mechanical tissues, provide structural support, while dermal and parenchymatous tissues increase water storage capacity and minimize water loss. High proportion of vascular tissues help for better conduction of water and nutrients from soil and provide mechanical strength to plant organs under stressful conditions. More importantly, increased concentration of organic osmolytes aid in turgor pressure maintenance, and ionic content (K^+^ and Ca^2+^) is crucial for better plant growth and development.

## Figures and Tables

**Figure 1 plants-13-00203-f001:**
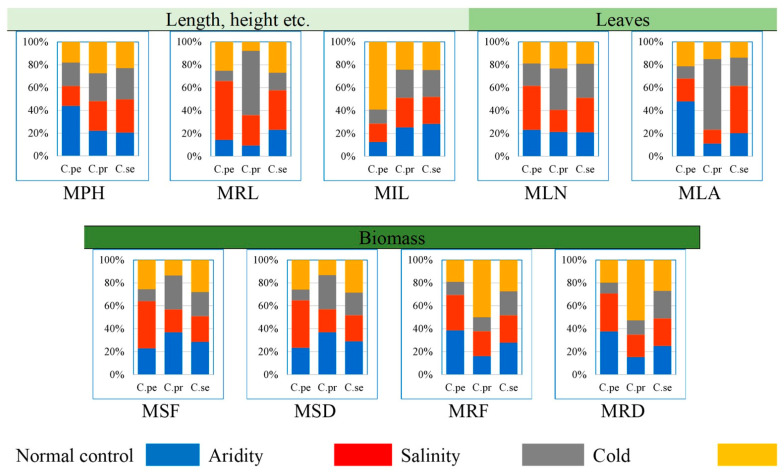
Proportion of morphological traits of three *Cenchrus* species under multiple abiotic stresses. Morphology: MPH—plant height (cm), MRL—root length (cm), MLN—number of leaves per plant, MIL—inflorescence length (cm), MSF—shoot fresh weight (g plant^−1^), MSD—shoot dry weight (g plant^−1^), MRF—root fresh weight (g plant^−1^), MRD—root dry weight (g plant^−1^), MLA—leaf area (cm^2^).

**Figure 2 plants-13-00203-f002:**
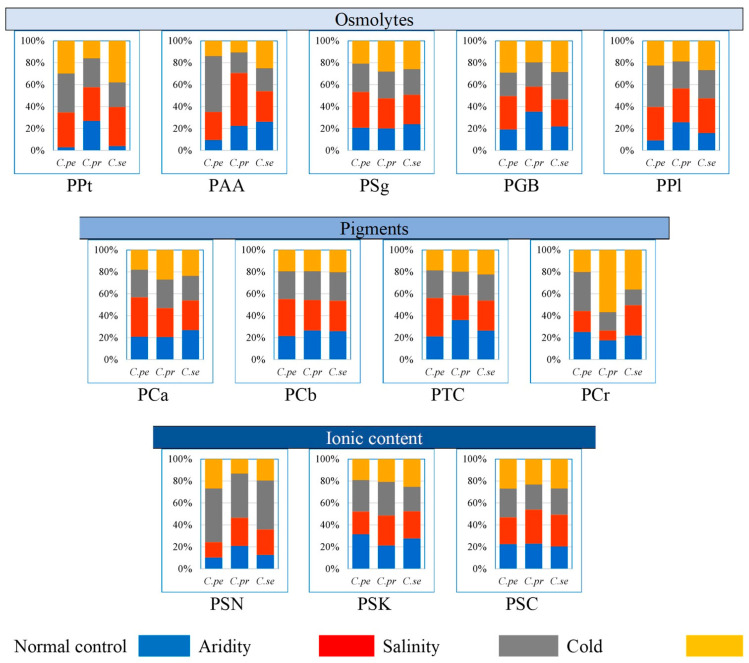
Proportion of physiological traits of three *Cenchrus* species under multiple abiotic stresses. Osmolytes: PPt—total soluble proteins (μg g^−1^ f w.), PAA—total free amino acids (μg g^−1^ f w.), PSg—total soluble sugars (mg g^−1^ d w.), PGB—glycine betaine (μmol g^−1^ f w.), PPl—proline (μmol g^−1^ f w.). Pigments: Pca—chlorophyll *a* (mg g^−1^ f w.), PCb—chlorophyll *b* (mg g^−1^ f w.), PCr—carotenoids (mg g^−1^ f w.). Ionic content: PTC—total chlorophyll (mg g^−1^ f w.), PSN—shoot Na^+^ (mg g^−1^ f.w.), PSK—shoot K^+^ (mg g^−1^ f.w.), PSC—shoot Ca^2+^ (mg g^−1^ f.w.).

**Figure 3 plants-13-00203-f003:**
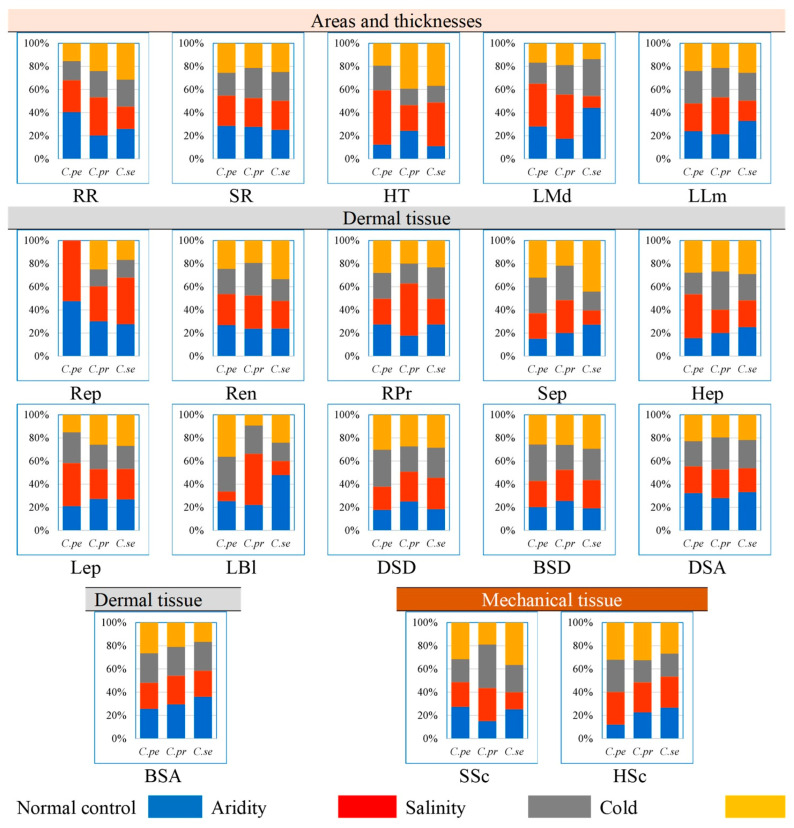
Proportion of areas, thicknesses, dermal, and mechanical tissue traits of three *Cenchrus* species under multiple abiotic stresses. Dermal tissue: Rep—root epidermal thickness (µm), Ren—root endodermal thickness (µm), RPr—root pericycle thickness (µm), Sep—stem epidermal thickness (µm), Hep—leaf sheath epidermal thickness (µm), Lep—leaf epidermal thickness (µm), LBl—leaf bulliform thickness (µm), DSD—adaxial stomatal density per mm^2^, BSD—abaxial stomatal density per mm^2^, DSA—adaxial stomatal area (µm), BSA—abaxial stomatal area (µm). Mechanical tissue: SSc—stem sclerenchymatous thickness (µm), HSc—leaf sheath sclerenchymatous thickness (µm). Thicknesses and areas: RR—root radius (µm), SR—stem radius (µm), HT—leaf sheath thickness (µm), LMd—leaf midrib thickness (µm), LLm—leaf lamina thickness (µm).

**Figure 4 plants-13-00203-f004:**
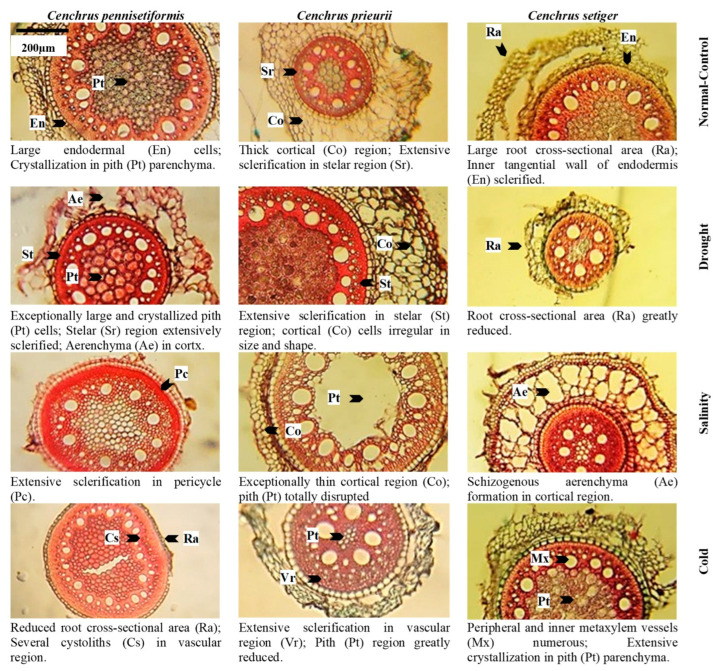
Root cross-sections of *Cenchrus* species under multiple abiotic stresses. Scale of measurement is given in the top left figure.

**Figure 5 plants-13-00203-f005:**
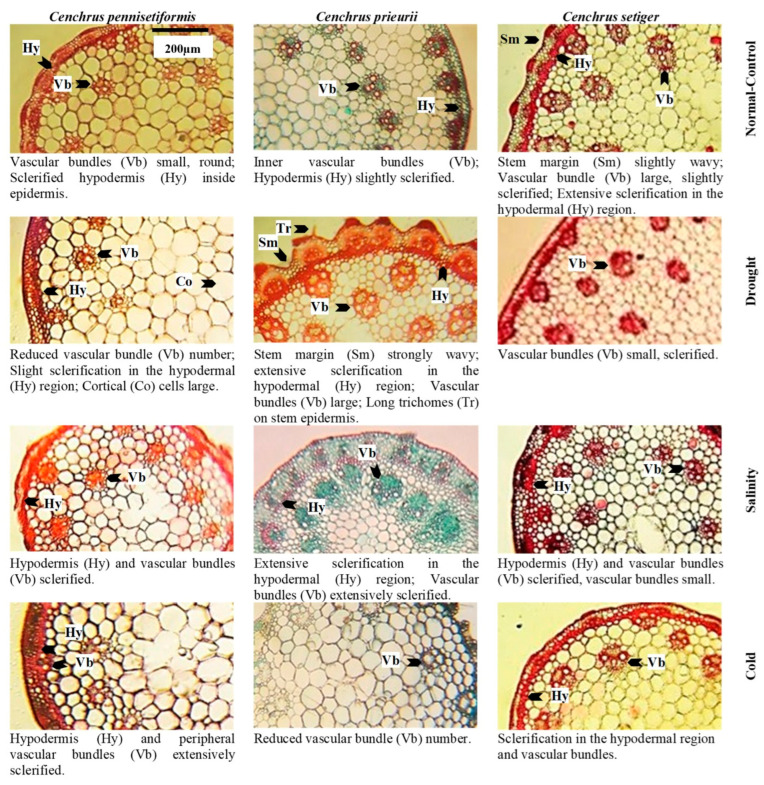
Stem cross-sections of *Cenchrus* species under multiple abiotic stresses. Scale of measurement is given in the top left figure.

**Figure 6 plants-13-00203-f006:**
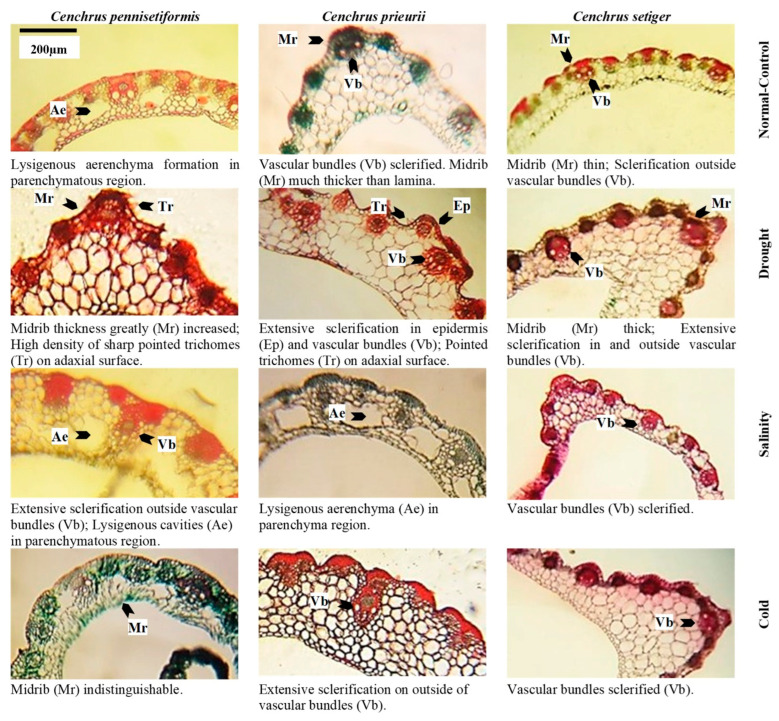
Leaf sheath cross-sections of *Cenchrus* species under multiple abiotic stresses. Scale of measurement is given in the top left figure.

**Figure 7 plants-13-00203-f007:**
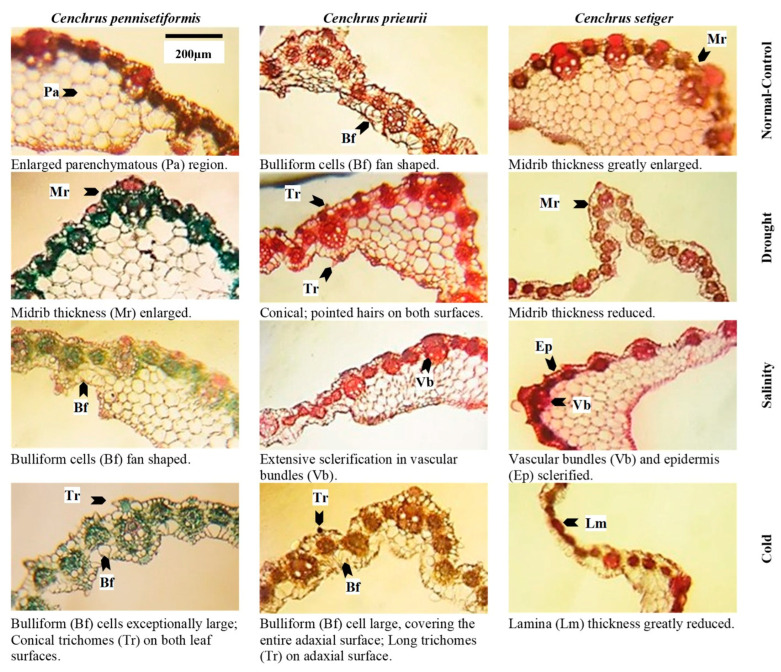
Leaf blade cross-sections of *Cenchrus* species under multiple abiotic stresses. Scale of measurement is given in the top left figure.

**Figure 8 plants-13-00203-f008:**
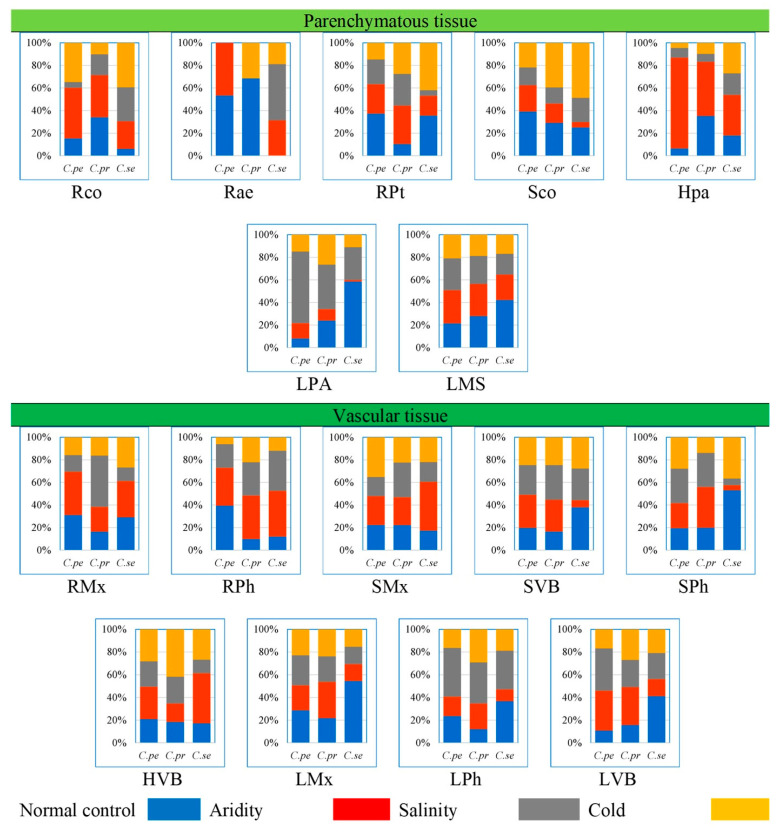
Proportion of parenchymatous and vascular tissue traits in three *Cenchrus* species under multiple abiotic stresses. Parenchymatous tissue: Rco—root cortical thickness (µm), Rae—root aerenchymatous area (µm^2^), RPt—root pith thickness (µm), Sco—stem cortical region thickness (µm), Hpa—leaf sheath parenchymatous cell area (µm^2^), Lpa—leaf parenchymatous thickness (µm^2^), LMs—leaf mesophyll thickness (µm). Vascular tissue: RMx—root metaxylem area (µm^2^), RPh—root phloem area (µm^2^), SMx—stem metaxylem area (µm^2^), SVB—stem vascular bundle area (µm^2^), SPh—stem phloem area (µm^2^), HVB—leaf sheath vascular bundle area (µm^2^), LMx—leaf metaxylem area (µm^2^), LPh—leaf phloem area (µm^2^), LVB—leaf vascular bundle area (µm^2^).

**Figure 9 plants-13-00203-f009:**
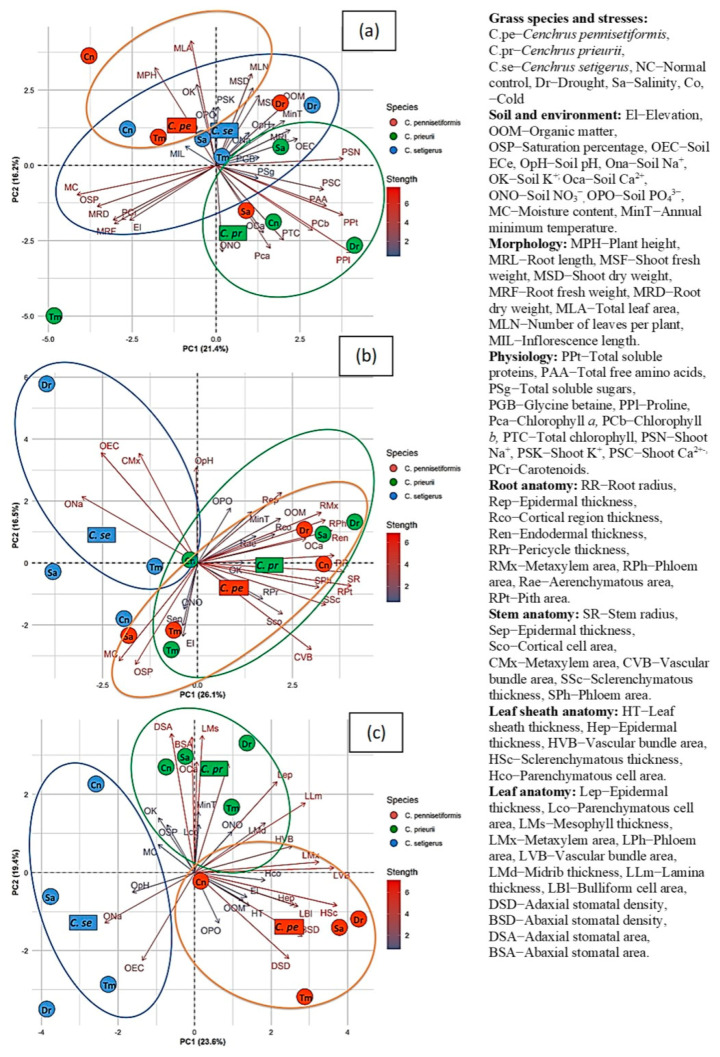
Principal component analysis showing relationship among soil/environment, morpholgy, and physiology (**a**), root and stem anatomy (**b**) and leaf sheath and leaf anatomy (**c**) of *Cenchrus* species under multiple abiotic stresses.

**Figure 10 plants-13-00203-f010:**
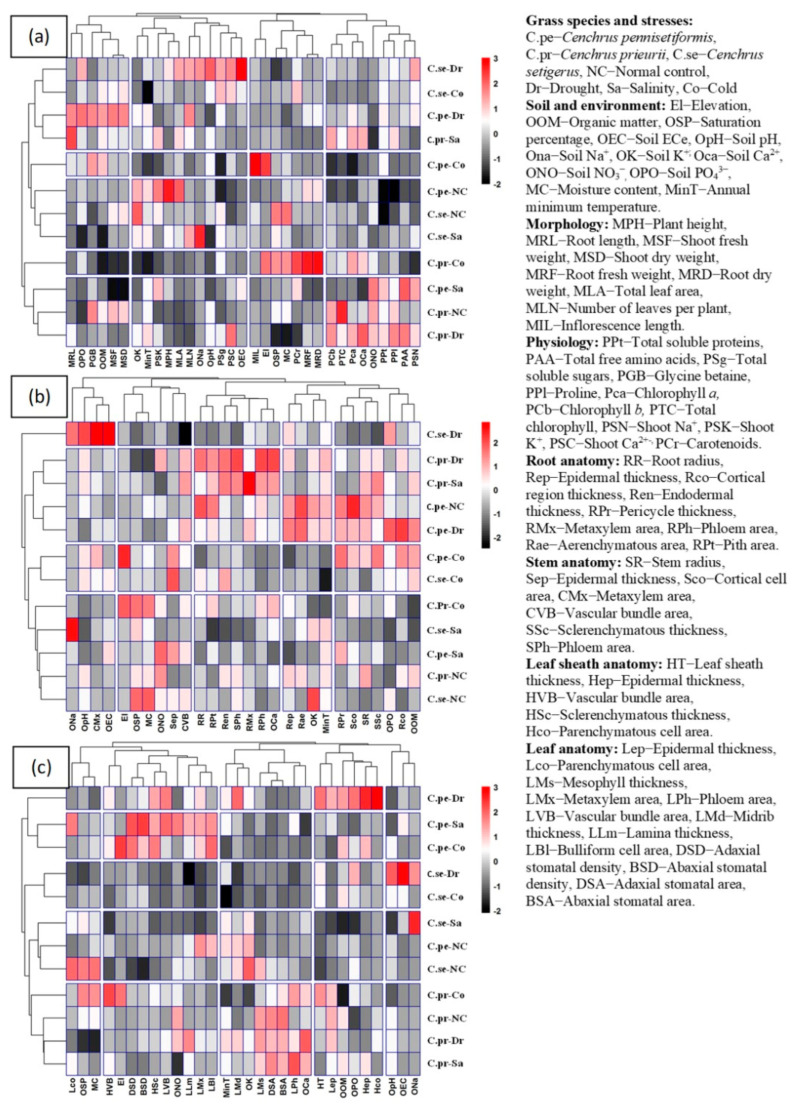
Clustered heatmap showing relationship among soil/environment, morphology, and physiology (**a**), root and stem anatomy (**b**) and leaf sheath and leaf anatomy (**c**) of *Cenchrus* species under multiple abiotic stresses.

**Figure 11 plants-13-00203-f011:**
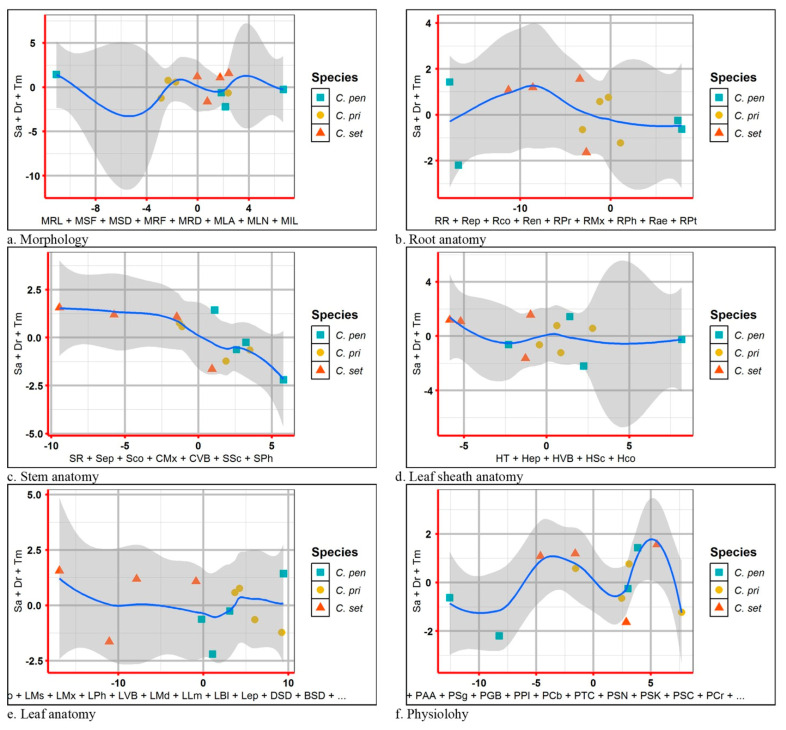
Estimated responses (**a**) morphological, (**b**) root anatomical, (**c**) stem anatomical, (**d**) leaf sheath anatomical, (**e**) leaf anatomical and (**f**) physiological traits of three *Cenchrus* species. Stresses: Sa—salinity, Dr—drought, Tm—cold. Morphology: MPH—plant height, MRL—root length, MSF—shoot fresh weight, MSD—shoot dry weight, MRF—root fresh weight, MRD—root dry weight, MLA—total leaf area, MLN—number of leaves per plant, MIL—inflorescence length. Physiology: PPt—total soluble proteins, PAA—total free amino acids, PSg—total soluble sugars, PGB—glycine betaine, PPl—proline, Pca—chlorophyll *a*, PCb—chlorophyll *b*, PTC—total chlorophyll, PSN—shoot Na^+^, PSK—shoot K^+^, PSC—shoot Ca^2+^, PCr—carotenoids. Root anatomy: RR—root radius, Rep—epidermal thickness, Rco—cortical region thickness, Ren—endodermal thickness, RPr—pericycle thickness, RMx—metaxylem area, RPh—phloem area, Rae—aerenchymatous area, RPt—pith area. Stem anatomy: SR—stem radius, Sep—epidermal thickness, Sco—cortical cell area, CMx—metaxylem area, CVB—vascular bundle area, SSc—sclerenchymatous thickness, SPh—phloem area. Leaf sheath anatomy: HT—leaf sheath thickness, Hep—epidermal thickness, HVB—vascular bundle area, HSc—sclerenchymatous thickness, Hco—parenchymatous cell area. Leaf anatomy: Lep—epidermal thickness, Lco—parenchymatous cell area, LMs—mesophyll thickness, LMx—metaxylem area, LPh—phloem area, LVB—vascular bundle area, LMd—midrib thickness, LLm—lamina thickness, LBl—bulliform cell area, DSD—adaxial stomatal density, BSD—abaxial stomatal density, DSA—adaxial stomatal area, BSA—abaxial stomatal area.

**Figure 12 plants-13-00203-f012:**
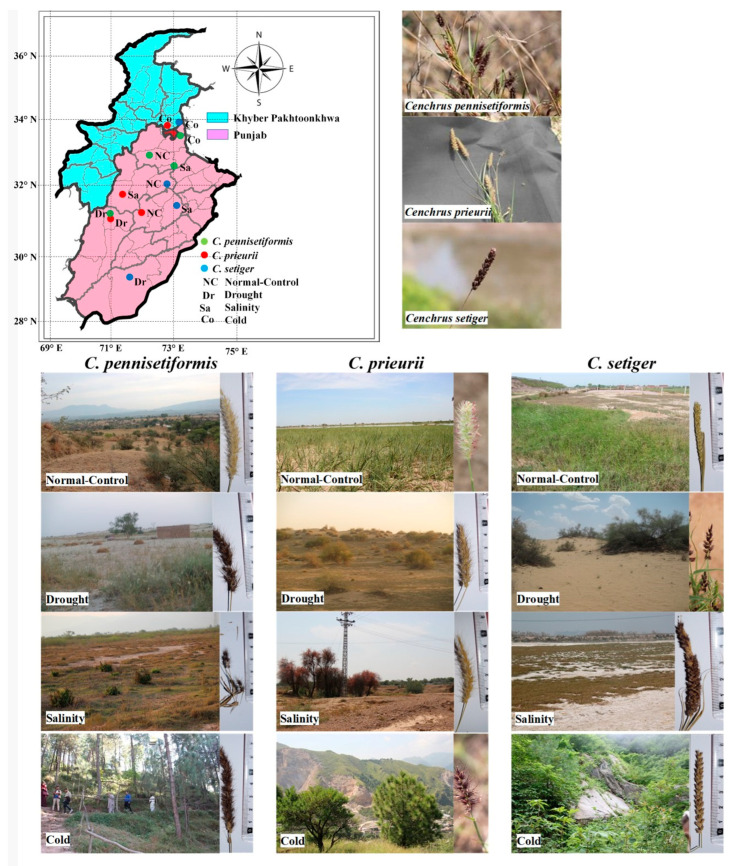
Map of the Punjab and Khyber Pakhtoonkhwa showing collection sites along with pictorial view of the habitats.

**Figure 13 plants-13-00203-f013:**
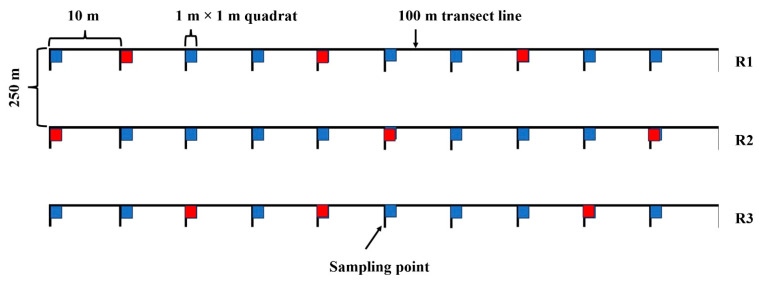
Sampling layout of *Cenchrus* spp. collection from areas exposed to multiple environmental stresses. Vegetation study was conducted from all quadrats (not given here), while red quadrats were selected for soil and plant analysis.

**Table 1 plants-13-00203-t001:** Environmental data of *Cenchrus* species collected from the Punjab and Khyber Pakhtoonkhwa.

	*Cenchrus pennisetiformis*				*Cenchrus prieurii*				*Cenchrus setiger*			
	Normal-Control	Aridity	Salinity	Cold	Normal-Control	Aridity	Salinity	Cold	Normal-Control	Aridity	Salinity	Cold
El	91.3	145.8	201.1	2182.7	147.3	151.4	292.5	2316.4	192.3	108.4	192.5	2446.3
ARF	324.3	122.4	309.4	1590.8	271.5	145.6	350.3	1603.7	368.5	117.6	320.7	1747.2
ASF	--	--	--	320	--	--	--	335	--	--	--	391
MxT	45.2	49.9	45.0	23.7	45.2	48	45.2	20.7	44.2	50.1	45	21.1
MnT	2.1	1.1	1.3	−3.1	1.1	1.2	1.1	−4.2	1.1	1.1	1.4	−5.6
Com	Cdd, Sba, Aja	Ajq, Lsc, Cpo	Fdi, Sva, Ala	Maf, Hna, Pro	Tdo, Ssp, Pka	Cpo, Cci, Aad	Cdd, Rst, Aad	Cde, Pwa, Tan	Cda, Cpo, Sba	Cdd, Hsa, Sim	Ala, Cla, Pka	Lpe, Acy, Pro
OOM	0.82	0.71	0.71	1.2	1.1	0.46	0.88	1.1	0.91	0.48	0.55	0.92
OSP	25.6	18.4	24.7	25.2	28.6	20.8	25.2	34.7	29.3	19.0	32.6	25.1
OEC	2.1	2.9	8.5	1.3	2.3	2.9	7.7	1.4	2.8	15.7	20.4	1.0
OpH	7.8	8.1	7.6	8.1	8.1	8.0	8.1	7.6	7.5	8.6	8.1	8.1
ONa	192.3	296.4	983.5	147.2	296.3	278.3	791.7	136.4	232.6	2117.1	2512.5	146.7
OK	83.7	112.1	67.4	76.0	87.4	52.3	141.3	101.5	127.5	147.4	75.1	77.5
OCa	57.5	78.9	46.6	53.6	67.4	95.1	82.2	77.5	67.2	57.6	78.2	49.9
OPO	6.8	10.3	4.9	7.1	4.2	5.6	7.2	7.3	7.5	9.1	3.8	7.2.6
ONO	3.5	4.3	4.8	2.7	4.4	4.1	1.8	3.5	2.3	2.8	2.9	2.9

Plant species: Aad—*Aristida adscensionis*; Acy—*Aristida cyanantha*; Aja—*Aerva javanica*; Ajq—*Acacia jacquemontii*; Ala—*Aeluropus lagopoides*; Cci—*Cenchrus ciliaris*; Cdd—*Capparis decidua*; Cde—*Cedrus deodara*; Cla—*Cyperus laevigatus*; Cpo—*Calotropis procera*; Fdi—*Fimbristylis dichotoma*; Hna—*Hedra nepalensis*; Has—*Haloxylon salicornicum*, Lpe—*Lolium perenne*; Lsc—*Lasiurus scindicus*; Msa—*Myrsine africana*; Pka—*Phragmites karka*, Pro—*Pinus roxburghii*; Pwa—*Pinus wallichiana*, Rst—*Rhazya stricta*; Sba—*Saccharum bengalense*; Sim—*Salsola imbricata*; Sva—*Suaeda vera*; Ssp—*Saccharum spontaneum*; Tan—*Themeda anathera*; Tdo—*Typha domingensis*. El—Elevation (m); ARF—annual rainfall (mm); ASF—annual snow fall (mm); MxT—absolute maximum temperature (°C); MnT—absolute minimum temperature (°C); Com—Vegetation community; OOM—organic matter (%); OSP—saturation percentage (%); OEC—soil ECe (dS m^−1^); OpH—soil Ph; Ona—Soil Na^+^ (mg g^−1^ d.w.); OK—soil K^+^ (mg g^−1^ d.w.); Oca—soil Ca^2+^ (mg g^−1^ d.w.); OPO—soil PO_4_^3−^; ONO—soil NO_3_^−^ (mg g^−1^ d.w.).

**Table 2 plants-13-00203-t002:** Pearson’s correlation coefficients among soil, structural, and functional traits of *Cenchrus* species collected from the Punjab and Khyber Pakhtoonkhwa.

Variables	Oel	OOM	OSP	OEC	OpH	Ona	OK	Oca	ONO	OPO
MPH	−0.268	−0.176	−0.227	0.160	0.083	0.258	0.334	−0.361	−0.148	0.053
MRL	−0.039	0.613	−0.334	−0.159	0.074	−0.280	0.100	0.429	−0.546	0.476
MSF	−0.310	0.709	0.047	−0.091	0.220	−0.107	0.378	−0.043	−0.426	0.371
MSD	−0.303	0.709	0.044	−0.057	0.260	−0.108	0.396	−0.095	−0.447	0.425
MRF	0.403	−0.351	0.357	−0.295	−0.267	−0.271	−0.098	0.339	0.033	0.274
MRD	0.425	−0.372	0.391	−0.257	−0.298	−0.173	−0.125	0.321	−0.063	0.290
MLA	−0.206	0.113	−0.285	0.230	0.349	0.274	0.339	−0.188	−0.562	0.353
MLN	−0.367	0.076	−0.086	0.425	−0.061	0.601	0.259	−0.107	−0.706	0.460
MIL	0.702	0.300	0.067	−0.150	0.239	−0.054	−0.136	−0.256	−0.233	0.078
RR	−0.291	0.023	−0.199	−0.402	−0.022	−0.364	0.410	0.563	0.007	0.043
Rep	−0.385	0.180	−0.173	0.021	0.142	−0.107	0.354	0.398	−0.040	0.406
Rco	0.132	0.719	−0.391	−0.139	−0.013	−0.159	−0.011	0.113	−0.295	0.309
Ren	−0.186	0.544	−0.410	−0.207	0.397	−0.542	−0.028	0.570	−0.058	0.089
RPr	0.210	0.286	−0.346	−0.232	−0.207	−0.305	0.111	−0.322	0.090	0.121
RMx	−0.218	0.439	−0.223	−0.130	0.264	−0.341	0.163	0.515	−0.444	0.425
RPh	−0.165	−0.101	−0.418	−0.113	−0.016	−0.090	0.079	0.854	−0.026	0.023
Rae	−0.287	0.199	−0.099	−0.132	−0.212	−0.003	0.316	−0.045	−0.179	0.231
RPt	−0.004	0.122	−0.221	−0.407	−0.067	−0.695	0.136	0.437	0.096	0.253
SR	0.125	0.507	−0.256	−0.521	0.007	−0.528	0.063	0.360	0.027	0.012
Sep	0.179	0.334	−0.012	−0.092	−0.109	−0.370	−0.151	−0.559	0.006	0.088
Sco	0.216	0.217	−0.052	−0.453	−0.274	−0.475	0.090	−0.196	0.043	0.144
CMx	0.079	0.309	−0.462	0.738	0.718	0.373	−0.237	−0.227	−0.426	0.539
CVB	0.202	0.095	0.206	−0.719	−0.531	−0.566	0.203	0.373	−0.011	−0.013
SSc	0.327	0.453	−0.278	−0.534	−0.033	−0.545	0.033	0.163	−0.269	0.091
SPh	−0.025	0.212	−0.108	−0.446	0.130	−0.588	0.098	0.631	0.269	−0.177
HT	0.188	0.199	−0.100	0.265	−0.130	−0.157	−0.538	0.132	−0.010	0.520
Hep	0.302	0.581	−0.036	−0.204	−0.286	−0.333	−0.093	−0.032	−0.470	0.605
HVB	0.635	0.028	0.102	−0.280	−0.235	−0.474	−0.523	0.318	0.066	0.257
HSc	0.539	0.296	−0.114	−0.211	−0.344	−0.434	−0.376	−0.166	0.206	0.173
Hco	−0.262	0.478	−0.252	0.041	−0.240	−0.126	0.157	0.128	−0.130	0.490
Lep	−0.042	0.159	−0.048	−0.258	−0.231	−0.475	−0.194	0.576	0.356	0.042
Lco	−0.104	−0.386	0.583	−0.237	−0.424	−0.071	0.246	−0.280	0.302	−0.316
LMs	−0.200	0.054	0.146	−0.299	0.133	−0.363	0.249	0.569	0.393	−0.173
LMx	0.161	0.202	−0.164	−0.432	−0.280	−0.571	0.155	−0.125	0.317	0.056
LPh	0.166	−0.311	0.195	−0.399	−0.265	−0.284	−0.137	0.638	0.068	−0.214
LVB	0.068	0.161	−0.173	−0.154	−0.520	−0.394	−0.144	−0.019	0.292	0.139
LMd	−0.227	0.147	−0.017	−0.487	−0.432	−0.263	0.709	0.211	−0.070	0.199
LLm	0.030	0.126	−0.152	−0.550	−0.307	−0.605	0.215	0.260	0.390	−0.179
LBl	0.350	0.219	−0.313	−0.258	−0.070	−0.369	−0.114	−0.296	0.281	−0.160
DSD	0.433	0.075	−0.259	0.136	−0.212	−0.078	−0.590	−0.472	0.292	−0.149
BSD	0.234	0.044	−0.301	0.050	−0.390	−0.136	−0.525	−0.346	0.398	−0.243
DSA	−0.102	−0.005	0.139	−0.374	0.185	−0.252	0.077	0.746	0.172	−0.333
BSA	0.024	−0.037	0.161	−0.364	0.123	−0.268	−0.040	0.713	0.305	−0.364
PPt	0.043	0.417	−0.539	0.326	0.077	−0.045	−0.639	0.119	0.119	−0.027
PAA	−0.423	−0.047	−0.436	0.332	−0.053	0.003	−0.068	−0.002	0.570	−0.230
PSg	−0.283	−0.245	0.050	0.530	0.082	0.403	−0.006	0.129	−0.247	0.347
PGB	0.169	0.700	−0.186	−0.193	−0.080	−0.304	−0.203	−0.154	0.061	0.057
PPl	−0.203	0.036	−0.362	0.308	0.152	0.126	−0.423	0.536	0.247	−0.256
Pca	−0.088	−0.094	0.061	−0.044	−0.060	−0.171	−0.083	0.778	0.000	0.210
PCb	−0.312	−0.005	−0.062	0.002	0.200	0.061	0.046	0.770	0.091	−0.189
PTC	−0.222	0.120	0.097	−0.099	0.139	−0.114	−0.101	0.518	0.317	−0.274
PSN	−0.590	0.040	−0.583	0.731	0.069	0.425	−0.188	−0.158	0.233	−0.054
PSK	−0.533	−0.075	−0.266	0.017	0.132	−0.160	0.403	−0.109	0.091	−0.056
PSC	−0.093	0.032	−0.673	0.650	0.445	0.350	−0.372	0.165	0.062	0.001
PCr	0.364	−0.400	0.371	0.168	−0.200	−0.158	−0.543	−0.329	0.174	0.203

Abbreviations: Soil and environment: El—Elevation, OOM—Organic matter, OSP—Saturation percentage, OEC—Soil ECe, OpH—Soil pH, Ona—Soil Na^+^, OK—Soil K^+^; Oca—Soil Ca^2+^, ONO—Soil NO_3_^−^, OPO—Soil PO_4_^3−^. Morphology: MPH—Plant height, MRL—Root length, MSF—Shoot fresh weight, MSD—Shoot dry weight, MRF—Root fresh weight, MRD—Root dry weight, MLA—Leaf area, MLN—Number of leaves per plant, MIL—Inflorescence length. Physiology: PPt—Total soluble proteins, PAA—Total free amino acids, PSg—Total soluble sugars, PGB—Glycine betaine, PPl—Proline, Pca—Chlorophyll *a*, PCb—Chlorophyll *b*, PTC—Total chlorophyll, PSN—Shoot Na^+^, PSK—Shoot K^+^, PSC—Shoot Ca^2+^, PCr—Carotenoids. Root anatomical traits: RR—Root radius, Rep—Epidermal thickness, Rco—Cortical thickness, Ren—Endodermal thickness, RPr—Pericycle thickness, RMx—Metaxylem area, RPh—Phloem area, Rae—Aerenchymatous area, RPt—Pith thickness. Stem anatomical traits: SR—Stem radius, Sep—Epidermal thickness, Sco—Cortical thickness, CMx—Metaxylem area, CVB—Vascular bundle area, SSc—Sclerenchymatous thickness, SPh—Phloem area. Leaf anatomical traits: Lep—Epidermal thickness, Lco—Parenchymatous cell area, LMs—Mesophyll thickness, LMx—Metaxylem area, LPh—Phloem area, LVB—Vascular bundle area, LMd—Midrib thickness, LLm—Lamina thickness, LBl—Bulliform thickness, DSD—Adaxial stomatal density, BSD—Abaxial stomatal density, DSA—Adaxial stomatal area, BSA—Abaxial stomatal srea. Leaf sheath anatomy: HT—Leaf sheath thickness, Hep—Epidermal thickness, HVB—Vascular bundle area, HSc—Sclerenchymatous thickness, Hco—Parenchymatous cell area.

**Table 3 plants-13-00203-t003:** Overall response of Cenchrus species exposed to multiple environmental stresses.

Traits	Normal control	Drought	Salinity	Cold
*Cenchrus pennisetiformis*				
Morphology	MPH, MLA, MRF	MRL, MLN, MSF, MSD, MPH	MRL, MLA, MSF, MSD, MRF, MRD	MIL
Physiology	PPt, PAA, PPl, PSN	PSg, Pca, PTC	PSg, Pca, PTC, PSN	
Stem anatomy	SCo		SMx	SSc
Leaf anatomy	LVB	Lep, LBl	Lpa, LPh	
Leaf sheath anatomy	HSc	HT, HPa		
Root anatomy	RR, RPt	Rep, RCo, RMx	RCo	RPh
*Cenchrus prieurii*				
Morphology	MSF, MSD		MRL, MLN, MLA, MRF, MRD	MRF, MRD
Physiology	PGB, PTC	PAA, PCr	PSN	PCr, PPt, PAA
Stem anatomy		SPh	SSc	SCo
Leaf anatomy		LBl. LMd, LLm, LMx, LVB, LPA	LPA, LPh	
Leaf sheath anatomy		Hpa	HSc, HT, Hpa	HT, HVB
Root anatomy	Rae, RPt, RPh	RPr, RPt, RR	RMx, Rep	RCo
*Cenchrus setiger*				
Morphology		MLA	MRL	
Physiology	PPt		PSN, PCr	PCr
Stem anatomy	SPh	SMx, Sep, SVB, SPh	SPh	Sep, SSc
Leaf anatomy	LPA, LMd, LMS, LMx, LVB, LBl	LBl, LMd, LPA, LPh, LVB		
Leaf sheath anatomy		HT, Hpa, HVB		HT
Root anatomy	Rco	RPh	Rae, RPt	Ren, Rco, RPt Rsp, RMx

Black font colour indicates the maximum value while red colour the minimum value. Morphology: MPH—plant height, MRL—root length, MSF—shoot fresh weight, MSD—shoot dry weight, MRF—root fresh weight, MRD—root dry weight, MLA—leaf area, MLN—number of leaves per plant, MIL—inflorescence length. Physiology: PPt—total soluble proteins, PAA—total free amino acids, PSg—total soluble sugars, PGB—glycine betaine, PPl—proline, Pca—chlorophyll *a*, PCb—chlorophyll *b*, PTC—total chlorophyll, PSN—shoot Na^+^, PSK—shoot K^+^, PSC—shoot Ca^2+^, PCr—carotenoids. Root anatomical traits: RR—root radius, Rep—epidermal thickness, Rco—cortical thickness, Ren—endodermal thickness, RPr—pericycle thickness, RMx—metaxylem area, RPh—phloem area, Rae—aerenchymatous area, RPt—pith thickness. Stem anatomical traits: SR—stem radius, Sep—epidermal thickness, Sco—cortical thickness, CMx—metaxylem area, CVB—vascular bundle area, SSc—sclerenchymatous thickness, SPh—phloem area. Leaf anatomical traits: Lep—epidermal thickness, Lco—parenchymatous cell area, LMs—mesophyll thickness, LMx—metaxylem area, LPh—phloem area, LVB—vascular bundle area, LMd—midrib thickness, LLm—lamina thickness, LBl—bulliform thickness, DSD—adaxial stomatal density, BSD—abaxial stomatal density, DSA—adaxial stomatal area, BSA—abaxial stomatal srea. Leaf sheath anatomy: HT—leaf sheath thickness, Hep—epidermal thickness, HVB—vascular bundle area, HSc—sclerenchymatous thickness, Hco—parenchymatous cell area.

## Data Availability

(1) The voucher specimens used for plant identification are deposited to the herbarium facility of the Department of Botany, University of Agriculture, Faisalabad, and are available for verification on request. (2) Anatomical slides, photographs, and raw data calculated from these photographs are available with the primary author and can be requested if needed.

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
