# Peer review of "Structural and Functional Strategies in Cenchrus Species to Combat Environmental Extremities Imposed by Multiple Abiotic Stresses"

_plants, 2024, doi:10.3390/plants13020203_

Round 1
Reviewer 1 Report
Comments and Suggestions for Authors
Manuscript ID: plants-2594197
"Structural and Functional Strategies in Cenchrus Species to Combat
Environmental Extremities Imposed by Multiple Abiotic Stresses"
Comments to the Author
Point 1: The Abstract is general and not informative. The Abstract needs modification. Moreover, the authors should focus on what was the problem, the hypothesis, the treatments, main results and conclusion. All abbreviations should be first identified before use them even if they were in abstract or another part of the manuscript
Point 2: The introduction does not point out the gap of the literature the study seeks to fill and novelty of the study over the existing literature. This point showed be further elaborated.
Point 3: The objectives and conclusion of the study are not clear and need to re-write
Point 4: A relevant hypothesis for the study is missing from the introduction. A true scientific question should be formed
Point 5: Deleted soil properties of the studied soil must be included
Point 6: The Discussion Section required substantial improvement. The discussed information should be separated from previously reported literature
Point 7: While your manuscript clearly reflects a sound understanding of the existing literature, I would like to emphasize the importance of including more recent publications in your references. Scholarly conversations are continually evolving, and including citations from the last three years (especially from 2022 and 2023) will ensure your work is positioned within the most current state of the field
Point 8: While your manuscript displays a strong command of the topic and presents compelling findings, I noticed some minor language issues and inconsistencies throughout the text. These could potentially hinder the clarity of your message and disrupt the reader's engagement with your work.
Point 9: References are not entirely correct, please check the Author's Submission guideline of MDPI
Comments on the Quality of English Language
Manuscript ID: plants-2594197
"Structural and Functional Strategies in Cenchrus Species to Combat
Environmental Extremities Imposed by Multiple Abiotic Stresses"
Comments to the Author
Point 1: The Abstract is general and not informative. The Abstract needs modification. Moreover, the authors should focus on what was the problem, the hypothesis, the treatments, main results and conclusion. All abbreviations should be first identified before use them even if they were in abstract or another part of the manuscript
Point 2: The introduction does not point out the gap of the literature the study seeks to fill and novelty of the study over the existing literature. This point showed be further elaborated.
Point 3: The objectives and conclusion of the study are not clear and need to re-write
Point 4: A relevant hypothesis for the study is missing from the introduction. A true scientific question should be formed
Point 5: Deleted soil properties of the studied soil must be included
Point 6: The Discussion Section required substantial improvement. The discussed information should be separated from previously reported literature
Point 7: While your manuscript clearly reflects a sound understanding of the existing literature, I would like to emphasize the importance of including more recent publications in your references. Scholarly conversations are continually evolving, and including citations from the last three years (especially from 2022 and 2023) will ensure your work is positioned within the most current state of the field
Point 8: While your manuscript displays a strong command of the topic and presents compelling findings, I noticed some minor language issues and inconsistencies throughout the text. These could potentially hinder the clarity of your message and disrupt the reader's engagement with your work.
Point 9: References are not entirely correct, please check the Author's Submission guideline of MDPI
Reviewer 2 Report
Comments and Suggestions for Authors
In this study, the author collected samples from different ecological regions in the Punjab and Khyber Pakhtoonkhwa to investigate structural and functional responses in these Cenchrus species (Cenchrus pennisetiformis, Cenchrus prieurii and Cenchrus setiger) to withstand to multiple abiotic stresses, i.e., drought, salinity and cold. The methods and results are acceptable. There are some essential problems should be addressed by authors, which are listed below.
1. There are some writing mistakes.
L32. Change “C pennisetiformis” to “C. pennisetiformis”.
L70, L372, L391, L433 and L724. There are obvious formatting errors.
L115. Change “NO₃⁻” to “NO₃⁻”, please check the full text carefully.
L147 L160. “chlorophyll a, b” or “chlorophyll a, b”, please write uniformly.
L212. Change “C, setiger” to “C. setiger”.
L224. “Fig 8” or “Fig. 4”, please write uniformly.
L234-L235. Change “C. pennise tiformis” to “C. pennisetiformis”.
L300. Change “p<0.05” to “p<0.05”, please check the full text carefully.
L724. Change “(Hitachi-220 Japan)” to “(Hitachi-220, Japan)”.
L29 L651. “Khyber Pakhtoonkha” or “Khyber Pakhtoonkhwa”, please write uniformly.
2. Suggest using the first letter of each word for abbreviations in the Figure annotation.
3. The first appearance of Latin requires a full name, while subsequent writing uses abbreviations to avoid repetition.
4. Multiple writing errors in the References. For example, Latin not italicized.
5. Some Figures should be modified.
In Figure 1. The species abbreviation should be italicized, please keep consistent with other Figures.
In Figure 9. Latin should be italicized.
In Figure 10. Please write “c.pr-Sa, C.Pr-Co and c.pe-NC” uniformly.
In Figure 11. Please keep the abbreviations of the species in the figure consistent with Figure 1.
Reviewer 3 Report
Comments and Suggestions for Authors
The manuscript 'Structural and Functional Strategies in Cenchrus Species to Combat Environmental Extremities Imposed by Multiple Abiotic Stresses' is providing a lot of information, both physiological and morphological regarding the adaptations of 3 species of Cenchrus to various environmental conditions.
It represents a large mass of data. However, the results might be presented in a more condensed way, especially due to the amount of abbreviations used and not usual that bring confusion. The recommendation would be to find a shorter way to present the most important data and put the rest as supplementary data. It is not clear for me how representative the figures are in terms of microscopic sections, nor about the variability when 'proportions' are expressed. Does this corresponds to one measurement? A range might be more appropriate than an average here, if it corresponds to an average. What about the biological variability?
The way the sampling is also raising some questions, when seeing the pictures of the environment, how can you be sure that you will find plants in the quadrats as it is presented here in 4.1.? In the figure 13, why is there 2 colors?
Regarding morphological data, in the present study, pictures of full plants or inflorescences of several individuals could give an idea of the biological variabilty.
Regarding the physiological traits 2.3., how many replicates have been used?
Regarding the statistics (and this type of statistics is not the one I use the most), did you try redundancy data analysis (RDA)?
Besides the above comments, I would maybe recommend that this type of research may be more suited to environmental or ecological journals.
Comments on the Quality of English Languageline 49: closure stomata should be stomatal closure
Line 67-68: a verb is missing
line 79: anjan ghas is probably anjan grass
Reviewer 4 Report
Comments and Suggestions for Authors
Overall statement:
The manuscript conducted a study on the structural and functional traits of Cenchrus species in different habitats, and it try to reveal the alteration modes of traits in the group and species for survival under different environmental stresses. Many and useful data were collected in this study, and the results can support the understanding of biological characteristics and further researches on these species, especially the plasticity of the trait responses to stressful factors. But some parts of the manuscript are not well executed, some parts are not very clear. Thus, I believe there are some points that need to be improved, and some parts are recommended to rewrite.
My main suggestions for improvement are:
Through the manuscript state the changes of traits under the stressful environment, environmental factors can independently and interactively affect the traits and performances of plant species, the effects of different factors are mixed in the introduction and results parts. In addition, we can not simply define a habitat as the cold one or drought one when multiple factors are considered. I recommend that more concise questions should be given in the introduction, and state the changes of traits from intra-specific variation and plasticity, the environmental factors should be comprehensively analyzed. For more, maybe some important traits can be selected to explore the changes trends, so the results and conclusion will be more concise and clear.
Lines 56, 68, 72. the states are not clear, which factor and the trend should be check.
Lines 103. the “ multiple abiotic stresses” seems to study the interactions of factors.
Lines 106-108. It is difficult to answer this question based on the results in this paper.
Please check other points, focus on the meanings and logical of states.
Comments on the Quality of English LanguageModerate editing of English language required
Author Response
Dear Sir/Madam
Please see the attached file.

Round 2
Reviewer 1 Report
Comments and Suggestions for Authors
Dear Editor: The authors did not make any changes that I requested of them. I think they uploaded the wrong file. Here I uploaded the old report again, which contains the required amendments.
For example: In my pervious report I asked the author that: A relevant hypothesis for the study is missing from the introduction. A true scientific question should be formed.
The authors placed this paragraph in the new version of the manuscript with a different background, as it is, without any change.
Therefore, I cannot accept the manuscript in its current form

Comments on the Quality of English LanguageAuthor Response
Please see the attached file.

Reviewer 2 Report
Comments and Suggestions for Authors
Nice work
Author Response
No comments from this reviewer for 2nd round. Therefore, first round comments and response file is attached here to see the option for Manuscript submission in the interface.

Reviewer 3 Report
Comments and Suggestions for Authors
I cannot see improvements in the new version of the manuscript, therefore you will find my comments in the table you sent with the answers to the comments, my new comments are in bold format.
In its current format, I cannot recommend the manuscript for publication.
Reviewer 3 (31 Aug 2023 09:42:55) |
Answers to the comments |
It represents a large mass of data. However, the results might be presented in a more condensed way, especially due to the amount of abbreviations used and not usual that bring confusion. The recommendation would be to find a shorter way to present the most important data and put the rest as supplementary data. It is not clear for me how representative the figures are in terms of microscopic sections, nor about the variability when 'proportions' are expressed. Does this corresponds to one measurement? A range might be more appropriate than an average here, if it corresponds to an average. What about the biological variability? Rev2: There is nothing that has been changed here, so I would not consider this as a reply to the reviewer.
|
We added a new supplementary file, in which measurement data along with standard errors are given. |
The way the sampling is also raising some questions, when seeing the pictures of the environment, how can you be sure that you will find plants in the quadrats as it is presented here in 4.1.? Rev2: OK, short and quick 'answer.'.. In the figure 13, why is there 2 colors? Rev2: And why 2 colors? Did I miss that? |
We laid quadrats in a pure community of each grass species. It was 90% chance of the existence of plants from the planned quadrats. If a plant was missing from the quadrat, the very next quadrat was selected for the data analysis. |
Regarding morphological data, in the present study, pictures of full plants or inflorescences of several individuals could give an idea of the biological variability. Rev2: I cannot find supplementary tables |
Data is presented in supplementary tables. |
Regarding the physiological traits 2.3., how many replicates have been used? Rev2: In the text it says: Ten quadrats (1 m x 1 m) were placed along a transect line (100 m) at each study site, and one plant within each quadrat plants were selected. Three transect lines (replications) were positioned at each habitat, each separated by 250 m. Data of ten plants were then averaged and used in the analyses of different traits (Fig. 13). Sorry, but this is not indicated unless you tell me that this correspond to the red color? This does not correspond to the answer, is it correct? |
We selected plants from 9 quadrates, i.e., 9 replicates, which is indicated in Fig. 13. |
Regarding the statistics (and this type of statistics is not the one I use the most), did you try redundancy data analysis (RDA)? Ok, I would not comment further on that then |
Principal component analysis is the better option for this kind of data, which is presented in Fig. 11. |
Besides the above comments, I would maybe recommend that this type of research may be more suited to environmental or ecological journals. ok |
The scope of Plants is ‘Plants is an international and multidisciplinary scientific open access journal that covers all key areas of plant science. It publishes review articles, regular research articles, communications, and short notes in the fields of structural, functional and experimental botany. In addition to fundamental disciplines such as morphology, systematics, physiology and ecology of plants, the journal welcomes all types of articles in the field of applied plant science”. Since our study is focused on morphology, anatomy, physiology and ecology of Cenchrus species, we feel that this falls in the scope of this journal. |
Comments on the Quality of English Language line 49: closure stomata should be stomatal closure |
Corrected |
Line 67-68: a verb is missing still not done: The immediate response of abiotic stresses in the shortage of water, i.e., physiological drought [17]. |
Revised the sentence. |
line 79: anjan ghas is probably anjan grass OK, thanks for the info, |
Ghas is the local name of grass. This is Urdu name, not an English name. |
Author Response
Please the attachment.

Reviewer 4 Report
Comments and Suggestions for Authors
This paper was improved and most match the review suggestions, except some limitations on environmental analysis and writing.
I suggest this paper should be published in plants.
Round 3
Reviewer 1 Report
Comments and Suggestions for Authors
Dear Editor Journal of plants MDPI
Manuscript ID: plants-2594197
I re-reviewed the manuscript “Structural and Functional Strategies in Cenchrus Species to Combat Environmental Extremities Imposed by Multiple Abiotic Stresses" again and the authors made all the amendments that I asked before so I think the manuscript is suitable for publishing
Regards
Comments on the Quality of English LanguageDear Editor Journal of plants MDPI
Manuscript ID: plants-2594197
I re-reviewed the manuscript “Structural and Functional Strategies in Cenchrus Species to Combat Environmental Extremities Imposed by Multiple Abiotic Stresses" again and the authors made all the amendments that I asked before so I think the manuscript is suitable for publishing
Regards
Author Response

(The authors gave the same response as above.)
